# Bacteriophages as Targeted Therapeutic Vehicles: Challenges and Opportunities

**DOI:** 10.3390/bioengineering12050469

**Published:** 2025-04-29

**Authors:** Srividhya Venkataraman, Mehdi Shahgolzari, Afagh Yavari, Kathleen Hefferon

**Affiliations:** 1Department of Cell & Systems Biology, University of Toronto, Toronto, ON M5S 3B2, Canada; 2Dental Research Center, Avicenna Institute of Clinical Sciences, Avicenna Health Research Institute, Hamadan University of Medical Sciences, Hamadan P.O. Box 6517838678, Iran; 3Department of Biology, Payame Noor University, Tehran P.O. Box 19395-3697, Iran

**Keywords:** phage display, phage therapy, infectious diseases, mAbs, cancer, theranostics, Alzhiemer’s disease, CRISPR-Cas9 system

## Abstract

Bacteriophages, with their distinctive ability to selectively target host bacteria, stand out as a compelling tool in the realm of drug and gene delivery. Their assembly from proteins and nucleic acids, coupled with their modifiable and biologically unique properties, enables them to serve as efficient and safe delivery systems. Unlike conventional nanocarriers, which face limitations such as non-specific targeting, cytotoxicity, and reduced transfection efficiency in vivo, engineered phages exhibit promising potential to overcome these hurdles and improve delivery outcomes. This review highlights the potential of bacteriophage-based systems as innovative and efficient systems for delivering therapeutic agents. It explores strategies for engineering bacteriophage, categorizes the principal types of phages employed for drug and gene delivery, and evaluates their applications in disease therapy. It provides intriguing details of the use of natural and engineered phages in the therapy of diseases such as cancer, bacterial and viral infections, veterinary diseases, and neurological disorders, as well as the use of phage display technology in generating monoclonal antibodies against various human diseases. Additionally, the use of CRISPR-Cas9 technology in generating genetically engineered phages is elucidated. Furthermore, it provides a critical analysis of the challenges and limitations associated with phage-based delivery systems, offering insights for overcoming these obstacles. By showcasing the advancements in phage engineering and their integration into nanotechnology, this study underscores the potential of bacteriophage-based delivery systems to revolutionize therapeutic approaches and inspire future innovations in medicine.

## 1. Introduction

Bacteriophages (phages) are viruses that infect bacteria and have an RNA or DNA genome enclosed within a proteinaceous capsid. They comprise over 1031 phage types, forming the majority of biological particles in the world and occurring wherever bacteria reside [1]. Inside the phage-infected bacterium, the phages replicate and lyse out of the bacterial cell, killing the latter while releasing newly multiplied phage particles. Phages are considered powerful antibacterial agents, particularly against multidrug-resistant bacteria. As with other viruses, phages recognize distinct molecules on the bacterial cell surface, such as flagella, lipopolysaccharide, surface sugars, and peptidoglycan that constitute the capsules, as well as slime layers [2]. The phage recognition molecules are diverse, and, therefore, most of the phages exhibit specificity to the targeted bacterial strain. Upon infection, the genome of the phage enters the cell and rapidly shuts down several of the host cell processes while beginning the reproduction of the virus particles [3]. After lysing the bacterial cell, the phages proceed to infect neighboring cells. Phages display enormous potential in therapy against bacteria when administered by themselves or in combination with antibiotics and vaccines. Phages show great potential as biological nanomaterials and have emerged as attractive candidates for novel DNA delivery. Additionally, phages can be produced and purified on a large scale in a facile and economical manner.

A recent review by Cui et al. in 2024 discusses the mechanisms of phage action and the use of bacteriophage therapy against bacteria and cancer [4]. However, a deeper dive into therapeutic applications would provide practical insights. While phage-based systems are touted as versatile, there is insufficient comparison with competing technologies like CRISPR-based antimicrobials or conventional antibiotic alternatives. This would help in understanding the unique position of phage-based systems in medicine. There is limited exploration of hybrid approaches, like phage-nanoparticle conjugates, which could bolster phage efficacy while addressing stability and delivery issues.

Another recent review by Wang et al., in 2024, discusses the engineering and application of phage-based delivery systems for drug and gene therapy, highlighting their versatility and potential in nanomedicine [5]. It reviews various modification strategies, including non-covalent, covalent, and genetic modifications, to enhance phage functionality as delivery vehicles. The study emphasizes the unique advantages of different phage species, such as T4 and M13, in targeted therapies for cancer and bacterial infections. Future research directions focus on overcoming safety concerns and improving the efficiency of phage delivery systems in clinical applications. However, beyond addressing immediate challenges, the article does not sufficiently examine the impacts of phage-based delivery systems, such as environmental concerns or resistance development in phages themselves.

The novelty of our current review is that we present the use of phage therapy in all of the applications stated in the above two articles, in addition to the use of phages in treating veterinary diseases, neurodegenerative diseases, and the production of mAbs using phage display technology. We also present an extensive coverage of phages in photothermal (PTT) and photodynamic therapy (PDT), cancer immunotherapy, combination therapy, phages as theragnostic and bioimaging agents, as well as phage–nanoparticle conjugates. The advantages and disadvantages of the use of phage-based therapy are also discussed, along with regulatory issues concerning their use in disease prophylaxis and therapeutics. Also, the use of the state-of-the-art CRISPR/Cas9 technology in developing genetically engineered phages is elucidated.

Through this review, we aim to bring forth the most recent, complete coverage of phage applications across diverse fields and provide a theoretical source for the latest literature on the rational design and advancement of major phage-derived delivery systems in the treatment and prophylaxis of various human and veterinary diseases. This broad scope of uses highlights the ingenuity of bacteriophages in addressing several medical challenges and substantiates the necessity for a comprehensive strategy towards phage-mediated drug development.

## 2. Advantages and Disadvantages of Bacteriophage-Based Therapeutics and Regulatory Issues

### 2.1. Advantages of Phage Therapy

Phages are advantageous due to the plenitude of their abundance in the environment and their versatility. Bacteria can develop resistance to phages; however, unlike conventional antibiotics, it is often facile to obtain new phages sourced from the environment. Furthermore, phages can undergo adaptability to resistance, by means of natural selection or through directed engineering [6]. Although the use of natural phages is favored, phages can be genetically engineered to make them more immunogenic, with a wider host range or ability to transport specialized payloads, such as CRISPR, to more efficiently destroy host bacteria. Among other favorable characteristics of phages is their capability to replicate at the infection site, thus perpetuating treatment where it is greatly necessary [7]. This enhanced phage density needs to be adequately potent so that the phage numbers can be viably sustained close to the site of the target bacteria in order to reach the required levels of bacterial killing [8].

Many decades of investigations on phage-based therapeutics, inclusive of contemporary clinical trials, have shown that there are no adverse side effects in human subjects [9]. Also, due to the high stability of phages, they can be stored at ambient temperature for several months [10]. Moreover, they can be subjected to storage in colder temperatures or with reagents that can augment the stability of phages in an aqueous suspension [11]. They can be stored by encapsulation, freeze-drying, and spray drying. Finally, stability of phages is accomplished wherein the phage titer does not notably diminish for many days, while, on the other hand, certain phages are intact and can maintain their stability for many years [11].

Yet another desirable quality of phages is their easy deliverability. Facile injections, such as intravenous, intramuscular, and intraperitoneal injections, have all been employed to successfully deliver phages in both humans and animals. These modes of injection provide an effective means of delivering phages to almost all tissues and organs and are notably better compared to oral delivery [12].

Phage therapy offers a promising alternative to antibiotics in addressing rising antibiotic resistance. Phages can precisely target harmful bacteria while preserving beneficial microbiota, crucial for gut health. Their self-replicating nature at infection sites enhances efficacy and reduces dosing frequency. Unlike antibiotics, phages can penetrate resistant biofilms and adapt to bacterial defenses, disrupting infections effectively. This co-evolutionary adaptability positions phages as a powerful tool against antibiotic-resistant infections, transforming modern therapeutic strategies [13]. Despite its potential, achieving broad-spectrum efficacy in phage therapy poses significant challenges, including the specificity of phages to particular bacterial strains and the need for precise engineering to address diverse infections. Recently, combining antibiotics with an expanding variety of phage cocktails has led to a significant decrease in bacterial density. Key strategies included selecting broad host range phages to ensure wider bacterial targeting, optimizing phage formulations for increased effectiveness, and combining phage therapy with last-resort antibiotics. These approaches aim to overcome resistance hurdles and pave the way for more effective therapeutic interventions against MDR pathogens [14].

### 2.2. Disadvantages and Challenges of Phage Therapy

Several phages have an exclusively lytic lifecycle, however, certain phages do not at once kill the host bacterium. These are called temperate phages, which integrate their respective genomes into the bacterial chromosome and become latent. Following this, upon stress to the host bacterial cell, they reactivate and enter the lytic growth cycle. Stresses may include radiation, UV light, heavy metals, and temperature. Temperate phages also grant immunity from additional infections by related phages and enable horizontal gene transfer that could lead to the dissemination of toxins and antibiotic resistance. Hence, only completely lytic phages are employed for therapeutic purposes [15].

Phage-enabled bacterial lysis could lead to endotoxin release from the bacterial cell, including those of Gram-negative bacteria. Phage candidates need to be screened for lysogeny-determining genes or toxins, or antibiotic resistance [11]. Authentic therapeutic phages must exhibit high potential to get to and kill their bacterial targets, combined with a low propensity to negatively modulate the microenvironments at the site of application. Such properties can be ensured insofar as they are obligately lytic and possess stability under normal storage settings and temperatures while being subject to standard safety and efficacy studies. Under ideal conditions, a given phage meant for therapeutic use must be fully sequenced in order to ensure the absence of toxins and other unfavorable genes [16,17].

The primary goal of phage elucidation is to exclude those phages displaying inefficient killing potential towards their respective bacterial therapeutic targets. Poor virulence could occur as a result of weak absorption, lowered potential to circumvent bacterial defense mechanisms, or inefficient replication properties [18]. Additionally, phages displaying inefficient pharmacokinetics due to lowered absorption, dissemination, and survival in situ are not favorable for therapeutic applications [18]. Further, therapeutic phages must display lowered potential for transfer of bacterial genes between different bacterial targets (transduction) [16,17].

Characterization of phages includes elucidation of virion morphology, protein profiling, and genotype explication, in addition to whole genome sequencing [17], which can render the costs for exhaustive phage determination prohibitive. The overall aim, hence, is the identification of phages displaying favorable primary pharmacodynamics (virulence against bacteria), lowered secondary pharmacodynamics (low propensity to harm patients), as well as efficient pharmacokinetics (capability to reach their bacterial targets in situ) [18]. Phages incapable of addressing the above criteria must not be used as therapeutics. At the least, this requires avoidance of temperate phages and full genome characterization of the phages to exclude the presence of any virulence factors.

None of the antimicrobials exhibiting selective toxicity will be effective against all microbial targets. Generally, the narrow host range of the phages poses an impediment to identifying the susceptibility of bacterial targets towards specific phages. Nevertheless, since phages can often be used along with other phages (phage cocktails) and antibacterial agents, the phage products’ lytic spectrum may be wider than the activity spectrum of individual phage strains [19,20]. However, even phage cocktails that are broadly acting are usually more selective in their activity spectrum than conventional narrow-spectrum antibiotics.

The unfamiliarity with the use of phages as antibacterial agents to the Western medical establishment could be a great challenge for phage therapy. Also, phages as antibacterial ‘viruses’ might be misconstrued by the general public as having properties equivalent to other viral pathogens that themselves cause disease in humans [21]. However, thus far, public resistance has not yet emerged, and it is perhaps fortuitous that these bacterial viruses are called phages instead.

Phage therapy, while innovative and promising, is indeed not immune to resistance issues. Similar to antibiotics, bacteria can adapt to evade phage attacks. This resistance can arise through various mechanisms, such as mutations in bacterial surface receptors that phages use to attach and infect cells. Additionally, bacteria can acquire advanced defense systems like CRISPR-Cas, enabling them to recognize and neutralize phage DNA, creating a significant challenge for the long-term efficacy of phage-based treatments [14,22]. Unlike antibiotic resistance, which is often permanent and transmissible, phage resistance can sometimes be circumvented or reversed. A key strategy to address this challenge is the deployment of phage cocktails—combinations of different phages targeting the same bacterial species but utilizing distinct receptors or action modes. This approach enhances efficacy and minimizes resistance development [23]. Phage cocktails attack bacteria from multiple angles, minimizing the chances of bacteria developing resistance to all phages in the mix. Genetic engineering offers another approach by modifying phages to alter their binding sites or bypass bacterial defenses, enabling them to infect resistant bacteria. While these strategies hold great promise, they are still under development and require further research to ensure their effectiveness and safety for clinical application [13].

Phage therapy faces hurdles due to the need for individualized approval for each bacterial strain and the absence of standardized regulatory frameworks [24]. Addressing these challenges requires the development of flexible, adaptive regulatory systems tailored to the unique nature of phage therapy. The large-scale production of phages presents significant challenges in maintaining purity, stability, and consistency, which can be both complex and costly [25]. To address these issues, advancements in biomanufacturing technologies are essential, enabling more efficient and precise production processes [26]. The host immune system can recognize and neutralize phages, diminishing their therapeutic efficacy [27]. To address this challenge, encapsulating phages or employing genetic modifications to alter their immunogenicity offers viable solutions. Effective phage therapy relies on accurately matching phages to their bacterial targets, a process that can be complex and time-consuming [28]. To address this challenge, expanding phage libraries to include a diverse range of phages with broad host specificities can improve the chances of finding an appropriate match [29]. Limited awareness among clinicians and patients regarding the advantages of phage therapy hinders its adoption [25]. Conducting targeted educational campaigns and outreach programs can foster understanding of its benefits and applications. The absence of standardized protocols for phage therapy in clinical settings poses a significant challenge to its broader implementation [30]. Addressing this requires the development of comprehensive clinical guidelines informed by the latest research and clinical trial outcomes. The high costs and limited availability of personalized phage therapy present significant barriers to its widespread adoption [31]. Scaling up production and investing in research to develop cost-effective manufacturing processes can reduce expenses and improve accessibility.

### 2.3. Regulatory Issues Associated with Phage Therapy

Phages are not categorized as living beings or as chemicals, thereby complicating their regulation. Therapeutic phage compositions are defined as compounded pharmaceutical preparations or industrially generated medicinal products [9]. Currently, natural phages or their products can be handled by pharmacists in the EU as active ingredients or raw materials, provided there is compliance with the respective European Directive requirements and provisions for medicinal compounds meant for human use. Established programs for practicing phage therapy now prevail in France, Belgium, Georgia, Poland, Sweden, and the USA. In Australia and Europe, collaborative ventures have been productive in formulating standard phage therapy schemes to enable therapeutic applications [32,33]. The UK recently proclaimed that it would start the use of phage therapy on a compassionate basis via the National Health Service. Phage clinical development regulation in the USA is practiced by the Office of Vaccines Research and Review (OVRR) in the FDA Center for Biologics Evaluation and Research, which regulates the purity, safety, consistency, and potency of manufactured phages [34]. As with all of the drugs regulated by the FDA, phage product licensure also necessitates that a given phage product has been successful in precluding, treating, curing, or mitigating diseases in humans.

In countries other than the EU, phage therapy is administered on benevolent grounds in the scenario where other modes of therapy have failed or if the disease status is imminently life-threatening [9]. In some other regions, there is a necessity for creating a distinct regulatory scheme to enable rapid supply of phage cocktails for personalized treatments based on the concept of quality by design (QbD), which is already in use for the generation of biopharmaceuticals and assimilates product and process quality subject to risk analysis [35]. Discerning patients’ requirements coupled with distinct quality and scientific properties of the phage product associated with its efficacy and safety are critical aspects of QbD. As antimicrobial agents, decisions pertaining to the regulation of clinical efficiency of phage therapy would initially be founded on procedures analogous to those of conventional antibiotics.

## 3. Bacteriophage-Based Therapeutic Strategies: Phage Display Technology, Biopanning, and Applications Using Phage Display Technology

### 3.1. Phage Display Technology

Foreign peptide display on the surface of phages without impacting the phage infectious process was initially described by George Smith in the year 1985 [36]. Ever since, phage display technology has been used in a major way for varied applications such as the determination of epitopes, enzyme substrate identification, drug discovery, and protein evolution [37]. The phage capsid protein is fused to an exogenous peptide, which is displayed on the phage surface to generate a combinatorial phage. Phage display facilitates a physical link between the DNA sequence and the protein/peptide sequence that enables quick separation depending on binding affinity towards a distinct target molecule and provides the capability to characterize the displayed proteins/peptides following selection of phages with favorable binding properties [38,39]. Greg Winter’s research group used this technology in therapeutic protein engineering, particularly in the discovery and generation of antibodies [40].

Typically, phage vectors used for phage display are filamentous phages such as M13, Fd, and f1 [41]. The M13 phage possesses a simple structure comprising a circular, single-stranded DNA genome enveloped by its outer protein shell called the capsid. The phage display platform based on M13 is capable of displaying folded proteins/peptides having disulfide bonds [42]. These proteins include functional antibody fragments, peptide inhibitors, and various enzymes [43]. The T7 phage species constitutes another phage vector with an icosahedral head as well as a short tail [44,45]. The T7 phage outer shell consists of the 10A and 10B coat proteins, and foreign peptide sequences are usually displayed as 10B capsid protein C-terminal fusions [46]. The T7 phage undergoes a lytic cycle, and, therefore, its reproduction and display are not based on secretion via the bacterial membrane [47].

Phage library construction is enabled by genetic engineering through which exogenous, random oligonucleotide fragments are inserted into the phage structural genes to facilitate its transcription as well as translation, and the respective foreign protein/peptide encoded by the exogenous gene is displayed on distinct sites of the capsid proteins of the phage. The phage library, therefore, consists of a mix of millions of phage particles, each of which displays a random and unique set of proteins/peptides [48]. Based on the size of the displayed antibody, peptide, or epitope and the antigen nature, two major categories of libraries have been engineered, namely, phage peptide libraries and phage antibody libraries.

### 3.2. Production of Monoclonal Antibodies (mAbs) Using Phage Display Technology

Phage display enables the rapid production of monoclonal antibodies by the display of antibody fragments on the bacteriophage surface, allowing for the segregation of high-affinity antibodies from large antibody libraries. For this, genes that code for antibody fragments such as Fabs or scFvs or even fully human monoclonal antibodies (mAbs) are introduced into the bacteriophage genome, after which these phages are used to generate a library of multiple antibody fragments that are displayed on the phage surface [49,50]. This results in the creation of a phage library containing phages, each of which expresses a different antibody fragment. Subsequently, this library is exposed to the antigen target, and phages that specifically bind to the antigen are “panned”. In the next step, these selected phages are amplified, thus enabling the isolation and identification of high-affinity antibodies.

Thus, phage display technology is a facile and quick method for the generation of antibodies against a diverse range of molecular targets, including peptides, proteins, and even carbohydrates. Diverse and large antibody libraries are created, enhancing the chances of finding antibodies with the requisite affinity and specificity. As this entire process is conducted in vitro, it dispenses with the requirement of animal immunization as well as the complexities involved in the conventional hybridoma technology. The phage display platform enables direct access to the gene sequence coding for the chosen antibody fragment and thus facilitates further production and engineering.

Phage display has opened the door to the discovery and development of therapeutic antibodies against various diseases such as cancer, infectious diseases, and autoimmune diseases. It has also facilitated the generation of highly sensitive and specific diagnostic reagents to detect antibodies and antigens. It is an inimitable tool to identify epitopes, discern immune responses, and study protein–ligand interactions. The popular anti-inflammatory monoclonal antibody, adalimumab (Humira), has been discovered using phage display technology, and several other therapeutic antibodies are currently under development.

The phage-produced scFv antibody was shown to be functional in a simultaneous blockade of EGFR and HER2, suggesting its potential as a promising candidate for targeted therapy against various EGFR-overexpressing tumors [51].

The technology of phage display, initially developed for peptide-directed evolution, has been widely used to discover completely human antibodies, as it offers several remarkable advantages. The excellence of the phage display technology has been demonstrated by several approved mAbs, inclusive of many of the foremost mAb drugs available in the market. Particularly, mAbs targeting antigens that are difficult to target have been generated using phage display platforms, aside from their ability to overcome the disadvantages inherent to in vivo antibody discovery strategies. Currently, the new era of phage display libraries is being optimized for the discovery and identification of mAbs possessing ‘drug-like’ properties [52]. Zhang, in 2023 [52], enlisted the mAbs generated by phage display technology that have been hitherto approved for use.

In recent times, random phage peptide libraries are increasingly being used extensively for the selection of peptides having affinity to distinct target molecules. Such libraries are constructed via the introduction of degenerate oligonucleotides into the phage genomes. Presently, peptides 6–43 amino acids long have been displayed successfully on the phage surface as capsid protein–peptide fusions [53].

### 3.3. Biopanning Strategy

Biopanning constitutes an evolutionary selection procedure wherein functional peptides having high specificity and affinity to specific targets can be selected from a random, large library consisting of up to 109 phage clones [41,52] (Figure 1). Firstly, a phage library that is customized is constructed for the display of favorable foreign peptides. In the next step, this phage library is allowed to interact with the desired target molecule, such as a protein, peptide, or cell, by incubation. During this step, billions of phages having randomly displayed peptides bind competitively to the respective target molecules, retaining potential peptides possessing stronger affinities to the targets. In the third step, unbound and weakly bound phages are removed using a wash buffer, following which competitive elution or a low pH buffer is used to enable elution of the strongly target-bound phages. Subsequently, the eluted phages are used to further infect new host bacteria to generate a more selective phage library to carry out the next biopanning cycle. Phages with high target affinities can be generated only with at least three to five rounds of biopanning. In each biopanning cycle, the efficiency and stringency of the phage selections are augmented by the increase in the number of washing steps and by the reduction of the number of target molecules. Particular care should be observed to preclude contamination with wild-type phages during the process of biopanning, since minute degrees of contamination can cause a major portion of the phage pool to become wild-type phages after three biopanning rounds. Following a final round of phage selection, the exogenous DNA cloned into the phage genome is sequenced. The resulting amino acid sequences are the encoded peptide ligand that interacts with the target molecule.

In vitro biopanning enables recognition of peptides interacting specifically with individual target molecules [41]. In vivo phage biopanning using live animals and even in human patients is aimed at generating peptides targeting organs or tissues under physiological conditions. While it is not exactly the same as selection in vitro, this in vivo selection methodology involves the intravenous systemic injection of the phage display library into the body, followed by a time period of circulation, after which the preferred tissue or organ is isolated and homogenized. Subsequently, the phage is extracted for downstream sequencing and identification of the peptide [54,55].

Peabody et al., in 2024, described a method for discovering potent, specific binding partners targeting defined motifs of the complementarity-determining regions of engineered, chimeric antibodies via affinity selection as well as counter-selection of antigenic epitopes displayed on MS2-derived VLPs [56]. Through this, they identified families of MS2 VLPs interacting with antibodies that display the CDRs coded by the germline precursor of the HIV-1 broadly neutralizing mAb. Specifically, they engineered MS2-based VLPs capable of displaying related peptide families showing preferential interaction with the IGHV1–2*02-coded antigen-interacting regions of the B cell receptor [56]. Frietze et al., in 2017 [57], described a strategy to map the repertoire of antibodies against dengue virus disease in humans using an antigen fragment library specific to dengue virus displayed on the VLPs of phage MS2, combined with deep sequence coupled biopanning. Employing an array of sera sourced from dengue virus patients having acute secondary infection, they panned a library of dengue virus antigen fragments displayed on the MS2 VLP surface and described a population of peptide epitopes that were affinity-selected using deep sequence analysis [57]. This demonstrated that, despite notable differences in individual responses, there were several epitopes in the non-structural protein1 and the envelope glycoprotein that were generally enriched. By means of this approach, they produced an elaborate map of linear dengue virus epitopes targeted by antibody reactions to secondary dengue virus infection, thus presenting a viable method that can be used to combat other infectious pathogens of interest.

### 3.4. Phage Cocktails

The phage cocktail is a mixture of different phages expected to suppress phage resistance and have a broader host range than a single phage. However, randomly mixed phage cocktails exhibit fluctuating bactericidal effects because of the interaction between cocktail component phages [58,59]. Besides the phage display technology, phages have been used for phage-based therapy in their natural or engineered forms. Phages are highly specific to the bacterial cell that they infect, rendering them a narrow-spectrum infectious agent. So, phage-based therapy does not perturb normal microbial flora and, therefore, is unlike broad-spectrum antibiotics that can lead to other complications, such as the emergence of pathogens, including *Clostridioides difficile*. Whereas narrow-spectrum therapies require precise diagnosis, cocktails of multiple phages enable a wider spectrum of antimicrobial activity against established pathogens [6]. Such cocktails of phages could exhibit activity against a variety of bacterial strains of the same species, resulting in the destruction of the target bacteria, which may make them more efficient compared to treatment with single lytic phages [11]. Schmerer et al., in 2014, showed that synergy can be attained when one infectious phage facilitates infection of the same bacterial cell by another phage [60]. Such synergistic phages could greatly enhance the generation of phage preparations towards therapeutic use due to their ability to increase clinical efficiency [11].

Under ideal conditions, phage therapy of a given patient needs an appropriate choice of phages as per their specificity, also called affinity [61]. It is essential that the identified bacteria are sensitive to the selected phage, without which phage-based therapeutics will be rendered ineffectual. A more viable strategy would be the use of cocktails of phages that will enhance the lytic spectrum of the phage in addition to providing combinatorial synergy or complementation, i.e., phages infecting phage-resistant strains that arise from infection of another phage in the mix [20]. In practical terms, it would be best to choose phages having lytic activity against broad-spectrum bacterial strains. Additionally, after the selection of the therapeutic phage, it is vital to study the multiplicity of infection (MOI), i.e., the ratio of phage infections per bacterial cell ascertained at the onset of phage therapy, as well as the MOI input, i.e., the number of phages provided per cell [9]. Another parameter constitutes the killing titer, which is the count of effective phage particles used as estimated by the phage counts based on plaque number, which can be applied to integrate its therapeutic use [62]. This will successfully provide therapeutic efficiency and favorable pharmacodynamics [18].

Many studies have addressed the multidrug-resistance (MDR) issue by employing strategies to improve the antimicrobial efficacy of phage therapy against MDR *Klebsiella pneumoniae* strains, which are notorious for their resistance to conventional antibiotics [14,59]. The emergence of multidrug-resistant *Klebsiella pneumoniae*, including carbapenem-resistant *K. pneumoniae* (CRKP), as one of the most common and notable superbugs, has long been a major threat to public health. An in vitro antibacterial activity assay demonstrated that the phage cocktail consisting of GZ7 and GZ9 effectively inhibited bacterial growth and suppressed the production of phage-resistant bacteria. The therapeutic effects of the GZ7 + GZ9 cocktail and GZ7 alone on mouse pulmonary infections were nearly equivalent. Therefore, phages GZ7 and GZ9 showed potential as alternatives to antibiotics for treating pneumonia caused by multidrug-resistant *K. pneumoniae* [63]. Canine otitis externa, characterized by the involvement of diverse bacterial species, notably, *Pseudomonas aeruginosa* and *Staphylococcus pseudintermedius*, necessitates antibiotic administration as the primary therapeutic approach; however, prolonged treatment often precipitates antibiotic resistance. Therefore, the application of bacteriophages as antimicrobial agents has been of interest recently. However, phage therapy has limitations; its efficacy depends on the lytic capacity of the phage and the emergence of phage resistance, which can be overcome by using phage cocktails. The therapeutic potential of the phage cocktail, consisting of Pseudomonas phage pPa_SNUABM_DT01 and Staphylococcus phage pSp_SNUABM-J, was evaluated [61]. The results suggest that administering a phage cocktail solution with additional components could make phage therapy a more efficient treatment for otitis externa in dogs.

### 3.5. Phage Encapsulation

Capsids, known for their substantial loading capacities, are capable of safeguarding therapeutic payloads while enhancing their bioavailability and stability. Phage capsid manipulations and encapsulation techniques allow phages to serve as effective carriers for therapeutic agents and their targeted delivery. These techniques include the engineering of phage genetic materials for payload insertion, surface conjugation techniques, and encapsulation of drugs within capsids. Additional methods, such as electroporation of phage capsids for efficient drug loading, adsorption through electrostatic or hydrophobic interactions, chemical synthesis during phage assembly, and heat- or pH-induced loading processes, enhance the versatility and efficacy of phage-based delivery systems in precision medicine [5,64,65]. Furthermore, by attaching multi-targeting ligands or functional moieties to the phage surfaces, these engineered systems can bind to multiple receptors, enhancing their specificity and versatility for therapeutic applications [64,66]. Bacteriophage-inspired drug delivery systems functioning as nanocarriers have been utilized across various fields, including cancer therapy, gene therapy, bacterial infection treatment, vaccination, and the detection of serological biomarkers [64].

Bacteriophage-mediated drug delivery systems provide numerous advantages, such as enhanced drug loading capacity, improved stability of therapeutic agents, precise site-specific targeting, minimized toxicity, the ability to work synergistically with antibodies, and the ability to boost therapeutic efficacy. Studies demonstrated that improved phage-antibiotic formulations, involving attachment of chloramphenicol to M13 phages and azithromycin to bacteriophage Qβ, enhanced drug-loading. These formulations also achieved precise targeting, stability, and efficacy improvements over the free drug [67,68].

### 3.6. Stability of Phages and Phage Encapsulation Using Liposomes and Polymeric Microparticles to Enhance Phage Stability and Efficacy

Knezevic et al., in 2011, explored the effects of various environmental conditions on adsorption and deactivation of three Siphoviridae phages (called J 1, 001A, and σ-1) and four *P. aeruginosa* phages (called δ) [69]. All the phages were totally inactivated following exposure to pH 1.5 for 30 min. Viability was maximum at pH 7. Notable susceptibility differences were observed for these phages at pH values 3, 5, and 9. Phages δ and σ-1 were less susceptible under low pH conditions compared to 001A and J-1. Survival capability at pH 9 was greater than at pH 5. Phage δ was significantly neutralized by rhamnose, mannose, alanine, glucose, and glucosamine [70]. All the phages failed to survive at working concentrations of 0.3% silver nitrate. Phages 001A and δ were found to be highly sensitive to exposure to povidone-iodine. Complete inactivation was observed under exposure to even the minimum concentration of 0.5% for 30 min. The gp181 structural peptidoglycan hydrolase of the Myoviridae bacteriophage ϕKZ that locally degrades the peptidoglycan layer of P. aeruginosa during infection showed a gradual loss of enzyme activity because of thermal inactivation [71]. At 25 °C, there was 100% activity, which went down to 0% when exposed to 70 °C for 60 min. Activity was <20% upon exposure to 90°C for 10 min. At pH 6.2, 25 °C, and an ionic strength of 140 mM, optimal enzyme activity was reported. Any change in ionic strength and pH in either direction resulted in the loss of activity.

Towards addressing issues with phage stability, active phages can be encapsulated within liposomes to achieve enhanced, efficient targeting and delivery by shielding phages from degradation by acids in the stomach [72,73], as well as immune clearance [74]. The efficiency of bacterial killing by liposomal phage compared to unencapsulated equivalents has become an unforeseen advantage [75]. Colom et al., in 2015, conducted pioneering studies wherein the phage cocktail composed of UAB_Phi20, UAB_ Phi87, and UAB_Phi78 was encapsulated into cationic liposomes to achieve oral delivery towards containing Salmonella infection within the GI tract by the use of a broiler model [72]. The encapsulated phages were found to be less susceptible to acid degradation and had extended retention time within the intestine as compared to their non-encapsulated counterparts. By the use of their model for in vivo infection, both the liposomal and free phage cocktail afforded similar protection levels against the colonization of Salmonella during phage therapy. Nevertheless, there was a relapse of Salmonella infection within 72 h after the stoppage of treatment with free phage, while the protective efficacy of liposomal phage prevailed for at least a week’s time in accordance with their extended retention period in the intestine. Otero et al., in 2019, demonstrated that oral administration of liposome-encapsulated phage led to dissemination into different organs [74]. They developed a co-culture model of HT29 and CaCo2 intestinal cells in vitro along with Raji-B lymphocytes in which they found the liposomes adhering to the surface of the cells as aggregates and embedded within the cell membrane and also internalized into cells, which could explain their augmented retention within the intestine as observed by Colom et al. in 2015 [72].

Several polymeric systems have also been shown to sustain the release of embedded phage particles to enhance the residence time at the infection site, towards the betterment of therapeutic effects. Polymeric biocompatible and/or biodegradable polymers, including natural polymers, such as pectin, chitosan, and alginate, as well as synthetic polymers, including Eudragit and (poly (lactic-co-glycolic acid)(PLGA)), have been employed to encapsulate, embed, conjugate, and adsorb phages. By virtue of being biocompatible and intrinsically non-degradable in mammalian systems, alginate is a favorable candidate to carry pH-sensitive therapeutics as it undergoes shrinkage in environments with low pH (stomach) while, at high pH (intestine), it swells or dissolves [76]. Wall et al., in 2010, showed that alginate microbeads embedded with phages controlled infections due to Salmonella in market-weight swine [77]. Both prophylactic and therapeutic effects were observed, reducing the colonization of Salmonella in the cecum, ileum, and tonsils [77]. Similar efficiency was obtained from the administration of pigs with the encapsulated phage cocktail through direct feeding [78]. This offered an effective and practical strategy to decrease the colonization as well as shedding of Salmonella in pigs and, thus, in other farming animals as well. Another biodegradable polymer, Poly lactic-co-glycolic acid (PLGA), has been approved by the FDA and extensively investigated as a phage delivery vehicle. Pseudomonas phages were loaded by adsorption, followed by incubation of the phage suspension with porous PLGA microparticles for 4 h under gentle shaking. A loss of ~0.5 log was reported following storage for 14 days at room temperature, and phages were found to be released in a sustained manner, having a burst release of ~15% within the first 5 min. Phage–PLGA microparticles, upon blending with lactose and delivery into mice, showed dispersion of the microparticles all over the lung, facilitating rescue of the mice from fatal pneumonia caused by the pathogen, *P. aeruginosa* [79].

### 3.7. Phage Engineering Using CRISPR-Cas9 Technology

Since phages are the most abundant form of life found on Earth, they have the potential to provide an unlimited resource for biomedical therapies. While phage therapy itself was conceived by Felix d’Herelle nearly a century ago, the recent appearance of antibiotic-resistant bacteria has forced us to explore phage-microbe interactions with renewed interest [80]. A major limitation of phage therapy is the fact that phages are highly specific with respect to their hosts and thus cannot infect all of the sub-strains of a particular pathogenic strain. Thus, one phage strain is unlikely to target a wide enough host range of sub-strains to be very effective. Improvements in synthetic biology technologies, such as the CRISPR-Cas9 system, have thus enabled the engineering of phage for specific purposes. Engineered phages could act as tools for diagnostics, pathogen control, and gene therapy. However, the use of genetically engineered phages for various applications remains in its infancy. At present, research studies have been limited to only a few phages and their host [81].

An important goal of engineering phage in medicine remains the expansion of the host range via genome editing. If successful, engineered phage therapy has great potential and could help the fight against drug-resistant bacterial pathogens, as well as modify the expression of genes of selected host bacteria. Phages can also be designed to express toxins that kill bacteria. All of these renewed traits can be realized via synthetic biology.

An important concern with producing engineered phages is how to isolate them from their wild-type counterparts after using CRISPR-Cas9. Researchers found that the use of sequence-specific RNA-guided nucleases designed to cleave wild-type phage nucleic acid but leave recombinant genomes intact was a viable approach [82]. This process, known as negative selection, has been utilized on the virulent phages of *Escherichia coli*, *Lactococcus*, and *Streptococcus*.

Another benefit of engineering phage is the easy adaptation of a process known as genome rebooting. Synthetic phage genomes can be rebooted in either cell-free systems or in transfected *E. coli* recipient cells (Figure 2) [83].

Examples of engineered phages are now abundant in the literature. Wang et al., in 2022, demonstrated the use of Cas12a, from an endonuclease family that is distinct from Cas9, to enhance homology directed repair efficiencies up to 3-fold in human cells [82].

Chen et al., in 2019, explored using CRISPR-Cas9 to edit the T7 phage and eliminate wild type from recombinants [84]. Similarly, the T4 phage system was used for phage therapy applications, such as multidrug-resistant bacteria, and as a diagnostic for foodborne pathogens [85]. Duong et al., in 2020, were able to achieve an editing rate of >99% for multiple genes [86]. Krishnamurthy et al., in 2016, used genome-wide screening to identify potential genes that would confer antibiotic resistance, thus providing insight into the molecular mechanisms involved in antibiotic resistance [87]. A phage that infects Klebsiella pneumoniae was genome-edited successfully, as demonstrated by introduced point mutations and deletion mutants [88].

Hoshiga et al. (2019) used engineered phage therapy as a method for prophylaxis of food poisoning caused by *Escherichia coli* O157:H7, an important food pathogen that is responsible for life-threatening bloody diarrhea [89]. The authors artificially expanded the natural, narrow host range of the phage and used T2 phage as a proof of concept for this plan. The T2 phage cannot infect *E. coli* O157:H7 strains. The authors were able to genome-edit T2 to be able to infect *E. coli* O157:H7 as efficiently as the natural pathogen strain PP01. On the same theme of expanding phage host specificity, Ali et al., in 2023, were able to engineer phage and produce phage cocktails with a broad spectrum of activities [90]. This procedure could enhance the efficacy of treatment [90]. Dedrick et al., in 2019, successfully used engineered phages for the therapy of a cystic fibrosis patient with a drug-resistant form of *Mycobacterium abscessus* [91]. However, caution must be exercised in using these phages.

In some other studies, bacteriophages were engineered using CRISPR-Cas13a towards targeting and degrading bacterial RNA, thus efficiently hindering bacterial replication and blocking gene expression [92,93,94]. Mitsunaka et al. (2022) designed a unique cell-free engineering and rebooting system for phages through the assembly of several phage genomes, including synthetic and natural ones [81]. They created phages that were biologically contained and proved to be equally effective as parent phages towards the treatment of lethal septic infections in vivo [81].

Finally, Cheng et al., in 2022, developed a ‘stepping-stone’ strategy that could enable phage genome synthesis, screening, and rebooting of 90 phages that infect popular pathogens within a single, user-friendly bacterial cell [83]. In this case, the authors were able to custom-design synthetic phage genomes and assemble them from smaller DNA fragments, using Listeria monocytogenes L-form bacteria for transfection [95]. The work described in this research will enable the development of synthetic phages that can target more pathogens. These examples illustrate the robustness of using genome-edited phages for a multitude of applications.

## 4. Phage Applications in Medicine

### 4.1. Bacterial Infections Therapy

Due to their high specificity to bacterial hosts, phages can target multidrug-resistant (MDR) bacteria without impacting beneficial microbiota. This is particularly important in the treatment of infections caused by debilitating pathogens such as carbapenem-resistant Enterobacteriaceae (CRE), methicillin-resistant *Staphylococcus aureus* (MRSA), and vancomycin-resistant *Enterococci* (VRE). A Belgian conglomerate of 35 hospitals spanning across 12 countries and 29 cities reported the outcomes of a clinical investigation involving 100 patients receiving personalized phage therapy [96]. Clinical advancement and elimination of the targeted bacteria were observed, respectively, in 77.2% and 61.3% of the infections, proving that bacteriophage therapy is efficacious against MDR bacteria (Table 1).

MRSA is a major cause of acquired infections in hospitals and is considered notorious due to its multiantibiotic resistance. Several investigations have proved phage efficacy in lysing strains of MRSA both in vivo and in vitro. In a murine model having wound infection, it was found that a phage cocktail efficaciously decreased MRSA colonization [97]. Presently, over 14 clinical studies on phage treatment of *Staphylococcus aureus* infections are ongoing or completed [4]. Several studies involving the use of phages against antibiotic-resistant *Pseudomonas aeruginosa* in respiratory tract infections and cystic fibrosis have currently reached completion or are ongoing. In immunocompromised subjects, VRE causes severe, debilitating infections. Phages against VRE have shown favorable outcomes in preclinical investigations. In murine models, it has been shown that targeted phages could greatly diminish VRE colonization, which highlighted their potential to treat gastrointestinal infections due to VRE [98,99,100]. Phages have been shown to control VRE within dental root canals and reduce VRE biofilms [98,99]. The phage vB_EfKS5 is capable of containing *Enterococcus faecalis* infections in food systems [100]. CRE also poses a major threat because of its resistance to some of the last-resort antibiotics, such as carbapenems. Chung et al., in 2023, showed the potential of bacteriophage therapy to target CRE strains [101].

Wound infections are the major cause of septic complications in patients having burns and augment burn-associated mortality and morbidity. Jault et al., in 2019, used 12 naturally lytic bacteriophages (PP1131) against *Pseudomonas aeruginosa* to demonstrate a decrease in bacterial burden within burn wounds at very low concentrations and at a slower pace compared to standard of care [102]. The high-yielding FG1m and FM4b lytic bacteriophages against *Salmonella* showed promise for industrial production [103]. Phage morphology, spectrum of lytic activity, and association with microbial cells, including reproductive efficacy, duration of latency, and adsorption rate, were analysed. By genome-wide sequencing, the taxonomy of these phages was determined. The phages were found to be devoid of any undesirable genes coding for adhesins, toxins, or invasins, and no “islands of pathogenicity” were identified. These phages were found to be thermo-tolerant and show stability at 4 ± 2 °C as well as at room temperature (25 ± 2 °C). Nevertheless, their lytic activity reduces with rising temperature in accordance with earlier studies [104]. Both the phages were also found to be chloroform-resistant and proved to be stable in buffers with pH ranging from 6.0 to 9.0.

Table 1 enlists findings from studies using phage therapy to treat infectious bacterial diseases.

**Table 1 bioengineering-12-00469-t001:** Some examples of successful studies using phage therapy to combat human bacterial infections.

Infection Type	Pathogens/Infections	Findings	Models Used	Ref.
**Skin**	Pyogenic infections (*E. coli*, *Proteus, S. aureus*, *Pseudomonas*, *Klebsiella*)	86% full recovery, 14% improvement	Human patients	[105]
	*S. aureus*, *E. coli*, *Streptococcus*, *Proteus*, *P. aeruginosa* in ulcers	70% healing, 23% bacterial reduction	Human patients	[106]
	*P. aeruginosa*, *Enterococcus*, *Staphylococcus* in diabetic ulcers	Infection alleviated, no MRSA infection	-	[107]
	*K. pneumoniae* in burn wounds	More effective than gentamycin and silver nitrate	Mouse models	[108]
	*A. baumannii*	Smaller, cleaner wounds	Balb/c mice	[109]
	*S. aureus* in eczema and acne vulgaris	Reduced symptoms, no harm to commensals	Human patients	[110]
**Oral**	*A. actinomycetemcomitans* in periodontitis	99% bacterial killing	In vitro studies	[111]
	*S. sobrinus*, *S. mutans* dental biofilms	Reduced biofilm severity, decreased caries	Sprague Dawley rats	[112]
	Endodontic infection (*E. faecalis*)	Degraded biofilm	Ex vivo models	[113]
**Gastrointestinal**	*C. difficile*	Symptom resolution, stool normalization	Human studies	[114]
	Diarrhea (*EPEC*)	Infection controlled	Balb/c mice	[115]
**Respiratory**	*K. pneumoniae*	Reduced inflammation, bacterial burden	Swiss Webster mice	[116]
	Chronic *P. aeruginosa* infection	70% bacteria cleared	Mouse models	[117]
**Urinary Tract**	*K. pneumoniae* UTI	Cured by phage-antibiotic combo	Human patients	[118]
**Eye**	*P. aeruginosa* keratitis	Preserved corneal integrity, reduced bacterial load	Murine models	[119]
**Ear**	Chronic *P. aeruginosa* otitis	Reduced bacterial counts, no adverse effects	Human patients	[120,121]
**Nasal**	Chronic rhinosinusitis (*S. aureus*)	20% favorable outcomes	Human patients	[122]
**Sepsis/Bacteremia**	*E. coli*	95–100% survival rates	Murine models	[123]
**Liver**	*Cytolytic E. faecalis*	Lytic phages attenuated ethanol-elicited liver disease	Humanized mouse models	[124]
**Orthopedic**	Multidrug-resistant *E. coli, E. faecalis* (VRE), *S. aureus* (MRSA) in osteoarticular infections	Controlled infections associated with implants	Human patients	[125]

Phage therapy has been combined with antibiotics, which show synergistic effects towards the treatment of drug-resistant infections. This strategy was employed in treating a patient having a lung infection due to multidrug-resistant *P. aeruginosa,* which led to a significant decrease in bacterial load and augmented positive clinical outcomes [126]. Another intriguing study by Fujiki et al., in 2024, showed that phages are capable of driving selection to restore sensitivity to antibiotics in *P. aeruginosa* through chromosomal deletions [127]. Antibiotic and phage combinations were used to treat MRSA infections associated with biofilms [97], wherein the combination therapy was more effective in the disruption of biofilms and killing of bacteria when compared to either treatment alone. Gordillo Altamirano et al., in 2022, showed that combination therapy using phage øFG02 and ceftazidime antibiotic restored the efficacy of the antibiotic against *Acinetobacter baumannii* [128].

Treatment of infections due to bacteria forming biofilms and spores poses tremendous challenges due to their resistance to traditional antibiotics. In such situations, alternative therapeutic measures are required to successfully eliminate the etiological agent. Phages, by virtue of their capability to infect and eliminate bacteria [129,130,131], can also undergo exponential replication and, therefore, constitute an important agent to combat pathogenic bacteria [132]. Wounds such as pressure sores and diabetic foot ulcers that are chronic often contain biofilms, impacting the healing process. Phages can disrupt these biofilms and promote wound healing [133]. Further, the combination of phages and antibiotics enhances efficacy against biofilms as observed in an in vitro model wherein bacterial load was reduced and wound healing was promoted [134]. Phage therapy is effective against biofilms caused by *Pseudomonas aeruginosa* in cystic fibrosis patients, as noted by the disruption of biofilms and the elimination of bacteria [135]. Tan et al., in 2021, showed that in a subject having chronic obstructive lung disease, personalized phage therapy was successful in treating *Acinetobacter baumannii* infection resistant to carbapenem, demonstrating the promise of tailored phage therapeutics in the treatment of severe infections [136].

The cysteine, histidine-dependent amido hydrolase/peptidase (CHAPK) enzyme derived from bacteriophage was shown to be capable of rapidly lysing many MRSA strains and both disrupted and precluded staphylococcal biofilm formation [137]. The staphylolytic action of this peptidase was shown in vivo in a mouse model with no adverse effects. This study proved the strong potential of CHAPK as an inimitable therapeutic candidate for treating staphylococcal infections, in addition to providing an insight into basic enzymatic mechanisms of peptidoglycan hydrolases containing the CHAP domain.

Medical equipment, such as heart valves, prosthetic joints, and catheters, contains biofilms that pose major risks owing to their resistance to standardized treatments. Mirzaie et al., in 2022, used a phage cocktail to control Proteus mirabilis surface colonization on catheters used in the treatment of urinary tract infections, which showed potential to preclude device-associated infections [138]. Plaque or dental biofilms are significant contributors to periodontal disease and dental caries. Kowalski et al., in 2022, analyzed the potential of phages in the therapy of periodontal infections and management of periodontal disease [139]. Another investigation by Chen et al., in 2021, highlighted the promise of bacteriophage therapy against periodontitis, proving that phages that target dental biofilms could greatly decrease the formation of biofilms and promote oral hygiene [140].

Intracellular bacterial pathogens, such as *Chlamydia trachomatis*, *Salmonella* spp., and *Mycobacterium tuberculosis,* reside within their host cells, rendering them formidable to target using conventional antibiotics. Phage therapy affords a propitious approach to combat these infections by using the capability of bacteriophages to cause infection and replicate inside host bacterial cells. *Mycobacterium tuberculosis* mainly infects macrophages, wherein it can persist and evade the immune responses of the host. Schmalstig et al. (2024) showed that mycobacteriophages are capable of infection and replication within these macrophages, which led to a significant decrease in bacterial load intracellularly [141]. Similarly, intracellular Salmonella have been effectively targeted by bacteriophages, wherein the latter have been shown to decrease the bacterial load. This makes these phages adjunctive therapies to augment the efficiency of the available antibiotics and diminish the emergence of resistance [142,143]. As observed in animal models, phages have been shown to penetrate and obliterate *Salmonella* inside infected tissues, affording the potential to treat persistent infections that are recalcitrant to traditional antibiotics [144]. This could be particularly favorable in eliminating *Salmonella* strains that are multidrug resistant, where therapeutic options are highly circumscribed [29]. The lytic phage ΦCPG1, which is specific to chlamydia, has emerged as a propitious candidate to treat *Chlamydia trachomatis* (CT) exhibiting antibiotic resistance [145]. ΦCPG1 has shown broad-spectrum inhibitory effects against all CT serotypes, efficiently disrupting CT infection stages and repressing bacterial growth, thus proving its potential as an inimitable therapeutic agent.

### 4.2. Phage Therapy Against Human Viral Diseases

Phages have been investigated as vaccine delivery vehicles against several life-threatening infectious viral diseases. Table 2 presents major examples of bacteriophage therapy used to treat viral infections. The study by Pan et al. addresses the limited immunogenicity of current H3N2 influenza subunit vaccines by developing a T4 phage-based nanovaccine displaying HA1 and M2e antigens on each phage particle. The nanovaccine showed strong immune responses, balanced Th1/Th2 activity, enhanced CD4+ and CD8+ T cell effects, and provided 100% protection in mice against H3N2 influenza. These findings highlight its potential as a robust subunit vaccine strategy for influenza and other viruses [146].

### 4.3. Phage Therapy in the Treatment of Veterinary Diseases

Zoonoses comprise infectious diseases that are transmitted indirectly or directly between animals and humans. Many major zoonotic pathogens can asymptomatically colonize farm animals, which may lead to food chain contamination, posing public health hazards. Additionally, carcasses have been routinely sampled by government authorities over the last two decades, wherein the increased emergence of antibiotic resistance among foodborne pathogens has been identified. If this trend prevails, antibiotics may turn out to be ineffective in eliminating such pathogens in due course, and alternate strategies such as phage therapy could become necessary. Towards this objective, several research groups have developed phage therapy for the treatment of veterinary diseases, some of whose examples have been listed in Table 3. The safety and toxicity of a veterinary disinfectant against Salmonella derived from three highly virulent bacteriophage strains were investigated using rats, Soviet chinchilla rabbits, and outbred white mice [152]. The tests showed the high safety levels of this bacteriophage-based disinfectant and were approved for use as a supplementary disinfectant against *Salmonella* in livestock and veterinary facilities.

### 4.4. Phage Technology in Neurotherapy

Alzheimer’s disease (AD) is the most prevalent causative factor of dementia. It is a progressive, heterogeneous neurodegenerative disease, the majority of which occurs in mid to late adulthood [166]. Aβ accumulation has evolved as the principal focus of AD pathophysiology [166]. The development of Aβ into senile plaques contributes to several deleterious neuronal effects [167]. Hence, the inhibition, deterrence, and even eradication of amyloid deposits in the brain of AD patients are propitious [167]. These various conformations of Aβ have been employed as selection baits in the treatment of AD through phage display technology. A highly distinct Aβ1–10 affinity peptide with the sequence PYRWQLWWHNWS was discovered using phage display [168], wherein the peptide could convert Aβ plaques into clusters of short fibrils. This peptide assuaged Aβ-elicited PC12 cellular viability and apoptosis and conferred protection against Aβ-generated learning and memory loss in rats.

D-peptide inhibitors were developed based on the technology of mirror-image phage display. From this, a potential and novel D-enantiomeric peptide D3 candidate of the sequence RPRTRLHTHRNR was discovered against monomers or small oligomers of Aβ42 [169]. The D3 peptide not only inhibited the aggregation of Aβ but additionally could redissolve pre-formed Aβ fibrils. Moreover, treatment with D3 rescued Aβ-elicited cytotoxicity within PC12 cells and greatly diminished inflammation as well as Aβ plaque load in Tg mice. D3, upon oral administration, augmented the cognitive efficacy of young and older Tg mice having AD, led to a notable decrease in the number of amyloid deposits, and reduced the accompanying inflammatory response [170,171]. Subsequent pharmacokinetic investigations showed that D3 possessed high stability against proteolysis, exhibited efficient penetration across the brain, and had excellent oral biocompatibility [172].

Similarly, phage libraries have been used to identify peptide inhibitors against Tau protein aggregation and other AD-associated molecules, in addition to recognizing the regulation of metal-elicited AD by peptide chelators. Yamaguchi et al., 2020, proved that peptides derived from phages can penetrate the blood–brain barrier (BBB) efficiently [173]. BBB cell models and cell membrane receptors are a direct and favorable strategy for biopanning. Nevertheless, in vitro conditions could be distinctly different from the complex brain that comprises the in vivo environment. In vivo phage display has been shown to be highly effective in the selection of phages having increased organ specificity following systemic injection [55]. Many peptides penetrating the BBB subsequent to in vivo selection have been recognized and used for targeted delivery of drugs against AD [174,175]. Phage display-based peptides have been used for developing novel biomarkers to enable in vitro prognosis of AD. Additionally, these peptides have been used as contrast agents specific to amyloid plaques to facilitate molecular imaging in vivo.

The poor diagnosis of AD is mainly due to the usually delayed disease prognosis. Therefore, early, efficient, and precise diagnosis based on AD biomarkers can be very critical in identifying disease-modifying therapeutics. Further, early diagnosis by blood tests could be less invasive, less costly, and provide better disease management. Phage display technology has been used to screen for antibodies or functional peptides present in the blood of healthy individuals and AD patients, respectively, and to design a blood biomarker diagnostic test based on the phage-displayed peptides, which would provide novel insights enabling early AD diagnosis. Additionally, new AD biomarkers, such as P-tau 217 [176] and P-tau181 [177], have emerged in recent years. The use of phage display technology to generate highly sensitive and high-affinity molecular probes in combination with several biosensors to accomplish rapid detection of AD in patients’ blood would demonstrate tremendous potential.

### 4.5. Phages and Cancer Therapy

Phages have diverse morphologies, including tailed, icosahedral, and filamentous phages [178]. Phage’s noninfectious nature, biocompatibility, biodegradability, and non-teratogenic properties significantly reduce their in vivo toxicity [179]. Phages can be genetically and physicochemically modified to produce modified phages [178]. The extent of phage modification depends on the availability of functional groups, their pKa values, and the solution’s conditions [178]. The abundance of nucleophilic functional groups in phages allows multiple amino acids to participate in chemical reactions, but this can lead to mixed reactions [178]. Unintentional modification of residues can reduce phage infectivity. Low-abundance amino acids, such as cysteine or unnatural amino acids, can be targeted for better control [180]. Researchers have developed the SpyPhage system, a method for engineering phages with a SpyTag moiety, allowing for rapid surface modification with therapeutic proteins fused with SpyCatcher. The SpyTag/SpyCatcher system is a technology for irreversible conjugation of recombinant proteins. The peptide SpyTag (13 amino acids) spontaneously reacts with the protein SpyCatcher (12.3 kDa) to form an intermolecular isopeptide bond between the pair. DNA sequence encoding either SpyTag or SpyCatcher can be recombinantly introduced into the phage DNA sequence encoding a protein of interest, forming a fusion protein. These fusion proteins can be covalently linked when mixed in a reaction through the SpyTag/SpyCatcher system. This could revolutionize phage therapies without the need for live bacteria or genetic alterations [181]. Phages can be produced efficiently by purifying them from virus-infected bacteria or using transgenic bacteria.

Tumors are a complex mixture of cellular and non-cellular compartments and their interactions, called the tumor microenvironment (TME). TME features include the enhanced permeability retention (EPR) effect, hypoglycemic acidic niche, dilated vasculature, and abnormal lymphatics. The TME acts as a barrier against current tumor treatment approaches, significantly influencing cancer development, progression, and therapy response. Phages can effectively target tumor barriers by improving therapeutic agent distribution into tumorous tissue and fine-tuning immunological responses. Studies have explored phages’ interactions with cancer cells (often via integrin receptors), wherein they affect the expression of genes [182]. Studies reveal that phages, specifically filamentous phages presenting the peptides VSSTQDFP and DGSIPWST, are specifically internalized by SKBR-3 breast cancer cells. The entry of these phages involves energy-dependent mechanisms, causing cell membrane changes and reorganization of actin cytoskeletons [183,184]. For example, phage MS2 significantly affects the expression of genes involved in LNCaP prostate epithelial cell proliferation and survival. These genes, including AKT, androgen receptor, integrin α5, MAPK1, STAT3, and peroxisome proliferator-activated receptor-γ coactivator 1α, are involved in normal cellular processes and tumor progression. MS2 significantly impairs LNCaP cells by altering the expression levels of cancer progression genes [185]. Phage’s tumor targeting property is influenced by their inherent properties, including nanoparticulate features, nano-engineering load, targetability, and inherent immune stimulatory ability. The development of tools to modify phages, genetically or chemically, combined with their structural flexibility, cargo capacity, ease of propagation, and overall safety in humans, has opened the door to a myriad of applications [186]. The phages such as the Ff, M13, fd, and f1 classes and the T4, T7, and Lambda classes have been used for phage display. The M13 filamentous phage showed higher efficacy in cancer treatment compared to the other phages [187]. For example, the M13 phage pVIII major coat protein, modified with cyclic RGD peptides, shows improved internalization efficiency into HeLa cells, potentially aiding in cancer therapy or diagnostics after further modification with drug molecules or contrast reagents [188]. The research suggests that multivalent phage libraries could expand the range of ligands facilitating cell entry, potentially impacting imaging, drug delivery, molecular monitoring, and cancer cell profiling [184,189]. Generally, current trends in phage-based tumor treatments include targeted delivery of therapeutic agents, tumor-targeted immunotherapies, and combinational therapies (Table 4).

#### 4.5.1. Phages in Targeted Drug Delivery

Maximizing drug dosage in cancer therapy leads to off-targeted administration, rapid clearance, high drug resistance, and recurrence, and may result in high toxicity in addition to limited clinical applicability [202]. Advanced drug delivery strategies can improve therapeutic outcomes by enabling intracellular and targeted delivery, reducing doses, and enhancing drug accumulation on the target. Drug-carrying phages are a novel platform for targeted anticancer therapy. This method is based on phages that have undergone chemical and genetic manipulation. Phages can exhibit ligands that confer host specificity thanks to genetic modification or chemical conjugation. Phages can be loaded with a sizable payload of therapeutic agents (small molecular nucleic acids, protein drugs). Phage nanomedicines that are targeted cause endocytosis, intracellular degradation, and drug release, and inhibit the growth of the target cells in vitro and in vivo compared to the corresponding free drug [203]. As a proof of concept, MS2 modified with the SP94 peptide can deliver chemotherapeutic drugs, siRNA cocktails, and protein toxins to human hepatocellular carcinoma (HCC). These modified VLPs have a 104-fold higher avidity for HCC than other cells and can deliver high concentrations of encapsidated cargo. P94-targeted VLPs selectively killed the Hep3B HCC cell line at drug concentrations < 1 nM, while SP94-targeted VLPs induced growth arrest and apoptosis of Hep3B at siRNA cocktail concentrations < 150 pM [204]. Bar et al. demonstrated a more than 1000-fold increase in the efficacy of hygromycin when delivered via phages, as compared to conventional drug treatment in vitro using human breast adenocarcinoma SKBR3 cells [191]. Additionally, Du et al. successfully coupled phages specifically targeting the human hepatocarcinoma cell line BEL-7402 with doxorubicin, resulting in a notable reduction in tumor growth and improved long-term survival in xenografted mice treated with drug-loaded phages, in comparison to free drug treatment [190]. In one study, Phage M13, a specific peptide, was found to be effective and specific against MCF-7 cells. When coupled with Tungsten disulfide (WS2), polyethylene glycol (PEG) and doxorubicin, it significantly improved therapeutic outcomes for chemotherapy [205]. The FA-M13-PCL-P2VP nanoassemblies, consisting of a shell modified M13 phage with folic acid (FA) and a core PCL-P2VP copolymer loaded with doxorubicin, were developed for drug protection and release, showing significantly higher tumor uptake and selectivity compared to free DOX [206]. The overexpression of major histocompatibility complex class I chain-related A (MICA) in cancer cells can be used to effectively deliver drugs to these cells, making it a useful targeted molecule. The 1-ethyl-3-[3-dimethylaminopropyl] carbodiimide (EDC) chemistry was employed to conjugate the anticancer drug DOX to the major coat g8p protein of M13 filamentous phages that carry anti-MICA antibodies. These drug-carrying phages specific to MICA antigens are more effective than free doxorubicin in killing cell lines expressing MICA [203]. The Salmonella typhimurium phage P22 virus-like particles (VLPs) have been modified to transport DOX due to their spacious interior cavity. These VLPs, composed of 420 coat proteins, target specific cells using affibody molecules. The modified P22 VLPs showed high cellular uptake in MDA-MB-468 and SK-BR-3 cells, overexpressing Epidermal growth factor receptor (EGFR) and HER2 [207,208,209]. Phage MS2 was used to deliver Tl+, an apoptosis-inducing agent, into tumor tissue. The iRGD peptide was conjugated to MS2 capsid proteins. Peptide-modified MS2 caused cell death in human breast cancer cells and necrosis in a mouse model [210]. Phage-like particles (PLPs) from phage lambda have pharmaceutical-grade properties and can be targeted by fluorescein-5-maleimide and trastuzumab. Trz-PLPs are internalized by HER2 overexpression in breast cancer cells, leading to increased intracellular concentrations and prolonged cell growth inhibition [197]. The blood–brain barrier, composed of cells with tight junctions, serves to prevent the entry of small (<400 Da) molecules by 98% and large molecules (>400 Da) by 100% [211]. By employing Trojan horse strategies, phages have been designed to transport drug cargos across the blood–brain barrier. An example of this is the conjugation of a cell-penetrating peptide from the Tat protein of human immunodeficiency virus type-1 to the exterior of P22 phage particles carrying the snail neuropeptide ziconotide in various in vitro blood–brain barrier models [212]. Apawu and colleagues achieved the crossing of the blood–brain barrier in rats by conjugating the synthetic peptide angiopep-2 to the capsid of MS2, containing an MRI detectable Mn^2+^ coordinated porphyrin ring [213]. Researchers have developed miniature chlorotoxin inho (CTX-inho) phage particles with a minimum length of 50 nm, capable of targeting GBM22 glioblastoma tumors in mice brains. These particles can accumulate in brain tumors and carry transcriptionally active cssDNA when delivered to GBM22 glioma cells in vitro. The ability to modulate capsid display, surface loading, phage length, and cssDNA gene content makes it an ideal delivery platform [214]. Suthiwangcharoen et al. present a nanosized delivery system for hydrophobic antitumor drugs, such as doxorubicin, encapsulated in poly (caprolactone-b-2-vinylpyridine) (PCL-P2VP) coated with folate-conjugated M13 (FA-M13). The nanoassemblies are stable at physiological pH but degrade at lower pH. DOX release is faster under acidic conditions, and the particles show greater cellular uptake and cytotoxicity against cancer cells [206]. To demonstrate the feasibility of M13 as a vehicle for drug delivery, doxorubicin (DOX) was loaded on the phage. To enable the targeting of tumors by M13, peptides were displayed on the gene-3 minor coat protein (protein-3, P3) or the gene-8 major coat protein (protein-8, P8). The fusion of the p3 phage coat protein with the SPARC Binding Peptide (SBP) sequence SPPTGIN was implemented. This fusion was carried out to exploit the properties of the Secreted Protein Acidic and Rich in Cysteine (SPARC), which is an anti-adhesive and promigratory matricellular glycoprotein that is abundantly expressed in aggressive forms of melanoma, breast, brain, prostate, colon, and lung cancers. As a drug delivery system, doxorubicin (DOX) was incorporated into the phage. For effective release of DOX, the p8 phage coat protein was linked to the peptide motif DFK, which is specifically recognized by cathepsin B, a lysosomal cysteine protease. This modification is expected to facilitate intracellular drug release, as cathepsin-B is overexpressed in prostate cancer. To attach DOX to the p8 aspartic acid residue, chemical coupling was employed. In addition, the p9 coat protein of M13 phage was enzymatically biotinylated and loaded with streptavidin-functionalized fluorophores for imaging (Figure 3) [215].

#### 4.5.2. Phages in Targeted Gene Therapy

Gene therapy has great potential to address genetic diseases through various technologies such as RNA interference, genome editing, and mRNA vaccines. However, obstacles such as rapid clearance, inadequate accumulation, and inefficient transfection efficiency need to be overcome. Bioinspired and biomimetic gene delivery systems have emerged, overcoming biological barriers and improving pharmacokinetic profiles. These advancements increase therapeutic effectiveness and minimize side effects, accelerating the clinical application of gene therapy [216]. As mentioned previously, peptides from phages theoretically allow the targeting of any organ or cell type and will then carry the nucleic acid therapeutics particle to the site of action. Larocca et al.’s research marked a milestone in phage-mediated genetic therapy, showing that a filamentous phage can selectively target and deliver a functional GFP reporter gene into mammalian cells [217]. Lankes and colleagues used an α V β 3 integrin-binding peptide on lambda phage to evaluate gene transfer in vivo. They found that administering mice with recombinant lambda phage virions containing luciferase enhanced gene transfer efficacy. Real-time imaging was used to analyze luciferase expression [218]. Bedi et al. used an MCF7-specific peptide for phage-mediated delivery of siRNA into cancer cells. They selected the peptide via biopanning and incubated it with GAPDH siRNA. This created nanophages, which protected siRNA from degradation and preserved its target cell-binding capacity. This study suggests that nanophages could be used as a peptide-targeted platform for therapeutic siRNA delivery into cancer cells [184,219]. Adeno-associated virus/phage (AAVP), a tumor-targeted phage with an adeno-associated virus genome, has shown effectiveness in delivering therapeutic genes to tumors in both laboratory and living organisms. However, obstacles hinder successful gene transduction [220]. TransPhage, a gene therapy vehicle, has shown excellent efficiency in transducing human cells, with up to 95% efficiency compared to adeno-associated virus vectors. In vitro, cancer cells expressing the membrane-bound fragment crystallizable (Fc) were effectively killed by CD16+ NK cells, and administration of the Fc gene significantly suppressed tumor growth [221]. Hajitou et al. developed a chimeric viral vector called AAVP, which inserts a chimeric genome containing an AAV transgene cassette into phage genomes, allowing for the delivery of siRNA instead of DNA [222]. Similarly, Przystal et al. have employed a related chimeric vector to deliver targeted suicide gene therapy to intracranial glioblastoma multiforme tumors in mice in order to inhibit tumor growth [223]. Qazi et al. used P22 VLPs as a programmable delivery vehicle for Cas9 and an sgRNA, isolating self-assembled VLPs from bacteria that encapsulated both proteins [224]. RNA nanotechnology offers a promising platform for creating tunable RNA-based modalities in tumor-bearing mice. The packaging RNA motif, derived from phage phi29 DNA packaging motor, can be used to fabricate RNA nanoparticles with targeting and therapeutic modules, potentially overcoming drug delivery barriers in cancer therapy [225]. Phage MS2 VLPs, coated with the TAT peptide, have been used to deliver microRNA (MiR-122) to target hepatocellular carcinoma (HCC). These VLPs inhibit insulin-like factor 1 receptor and cyclin G1, blocking carcinogenesis and promoting apoptosis in HCC cell lines [209,226]. The study of Sharifi et al. suggests a method for targeting EGFR-expressing cells using phage particles with EGF and GFP as tumor-targeting elements. The sfGFP-EGF coding sequence was inserted at the pIII gene in pIT2 phagemid, indicating potential for gene delivery and tumor detection [227]. The induction of apoptosis and inhibition of cancer cell proliferation is achieved through the reintroduction of maternally expressed gene 3 (MEG3) long non-coding RNA (lncRNA) in cancer cells (Figure 4) [228].

#### 4.5.3. Phages in PTT and PDT

PTT is a highly effective form of hyperthermia due to its safety and externally controlled specificity. It uses a photothermal agent (PTA) to convert electromagnetic radiation into thermal energy, preventing toxic effects on cancerous cells. Near-infrared (NIR) wavelengths are preferred for excitation, making PTT a highly effective method for treating cancerous cells [229]. For example, T7 phages with RGD motif affinity to human transferrin have been modified with Au nanoparticles (AuNPs), forming GP-phage-AuNPs. These nanoparticles have the ability to convert light into heat, making them effective heat sources for cancer cell damage. GP–phage–AuNP formulations rapidly killed prostate cancer cells under low light irradiation, while citrate-stabilized AuNPs and nontargeted AuNP clusters caused few cell deaths [85,230]. Researchers have developed a new strategy for breast cancer precision medicine using phage display techniques. They identified an MCF-7 breast tumor-targeting peptide and conjugated it to gold nanorods, enhancing cancer-killing efficacy. The peptides guide the drug to tumors without knowing the exact receptors, requiring less effort to explore patient-specific targeting molecules. By conjugating the peptides with AuNRs, the nanomedicine’s accumulation inside tumors improves cancer-killing efficiency. This approach could lead to highly efficient cancer treatment through PTT [231]. Bacteriophage Qβ’s self-adjuvating and site-specific functionalizability has led to the development of PhotoPhage, a photothermal, mildly immunogenic phage for effective photo-immunotherapy in triple-negative breast cancer models (Figure 5A) [232].

PDT is a method where a photoactive chemical, known as a photosensitizer, absorbs light to produce cytotoxic singlet oxygen. In cancer treatment, photosensitizers must accumulate within a tumor and be exposed to specific light wavelengths. This oxygen causes cell death or tissue destruction through processes such as apoptosis and necrosis [233]. Phages have been designed to transport photosensitizers to cancer cells, allowing targeted eradication through light activation [233]. Bacterial phage MS2 is a targeted, multivalent PDT vector for treating Jurkat leukemia T cells, selectively targeting and killing over 76% of cells 20 min post PDT exposure [234]. Recently, a T4 phage-based self-oxygenating nanoplatform, as a super tumor phage, has been used for improving PDT. The catalase (Cat) protein displayed on the capsid was applied to trigger hydrogen peroxide (H_2_O_2_) degradation. The 852 Cat molecules displayed like a brush on the phage surface, increase the oxygen concentration to 21.7 mg/L in a short time (1 min), which effectively relieves tumor hypoxia [235]. In another study, for targeting SKBR-3 breast cancer cells, the SKBR-3 phages were partially modified with a photosensitizer, pyropheophorbide-a (PPa), to create the phage-PPa complex that selectively kills cancer cells using PDT. This is achieved through the specific binding of PPPa to SKBR-3 cells and the subsequent selective killing of SKBR-3 breast cancer cells upon exposure to red light at 658 nm [233]. A study used nanoarchitectonics to design M13 phages as targeted carriers for eliminating cancer cells through photodynamic means. The phages were genetically refactored to present a peptide (SYPIPDT) that binds to the epidermal growth factor receptor (EGFR). These refactored phages were successfully internalized by A431 cancer cells overexpressing EGFR. The phages were then chemically modified to attach Rose Bengal photosensitizing molecules on the capsid surface, preserving the specific recognition of the SYPIPDT peptides. The M13EGFR-RB derivatives generated reactive oxygen species intracellularly, activated by ultralow intensity white light irradiation. The cytotoxic effect was observed at picomolar concentrations of the M13EGFR phage [236]. The study uses M13 phage as a targeted vector for efficient photodynamic killing of SKOV3 and COV362 cells. The phage is refactored to display an EGFR binding peptide, which is often overexpressed in ovarian cancer. When conjugated with chlorin e6 (Ce6), the new platform generates reactive oxygen species (ROS) and shows activity in killing these cells even at concentrations where Ce6 alone is ineffective [237]. Recently, a T4-Ce6-catalase (Cat)-lactate oxidase (Lox) (TCCL) biomimetic peroxisome was developed using T4 phage display technology for enhanced PDT, achieving high ROS generation efficiency and tumor inhibition rates, with low immunogenicity [238]. The synthesis of nanozyme-coated tumor-homing fd phage nanofibers using dual-peptide-displayed Pt-binding tumor-homing phage as a template was demonstrated. The nanofibers were genetically engineered to fuse Pt-binding peptides to pVIII and AR, forming fd-TN and fd-AR-TN. ICG photosensitizers were conjugated onto the nanofibers, forming ICG-loaded PtNE-coated tumor-homing phage. These nanofibers were injected into MCF-7 tumor-bearing mice for tumor therapy. The nanofibers catalyzed H_2_O_2_ decomposition and generated O_2_, enhancing the PDT of breast cancer [239].

Recently, a synthetic peptide, NW, has been developed using phage display technology to bind to M1 and M2 macrophages with high affinity. The peptide library was affinity selected on M2 macrophages blocked with NW peptide, resulting in peptides that bind to M2 but not M1 macrophages. The peptides were conjugated to the photosensitizer IR700 for cancer photoimmunotherapy. The results showed that M2 macrophages can be selectively targeted by the wild-type M13 phage, offering potential benefits for cancer treatment [240]. The study explores the use of phage nanofibers for targeted photodynamic cancer therapy. The researchers constructed a dually functionalized phage by partially conjugating pyropheophorbid-a onto a genetically engineered phage. The phage surface displays the peptide VSSTQDF, which is then used for targeted PDT. The cells were treated with the phage and irradiated using a cell viability kit. The results showed promising results in treating MCF-7 and target SKBR-3 cells (Figure 5B) [241].

**Figure 5 bioengineering-12-00469-f005:**
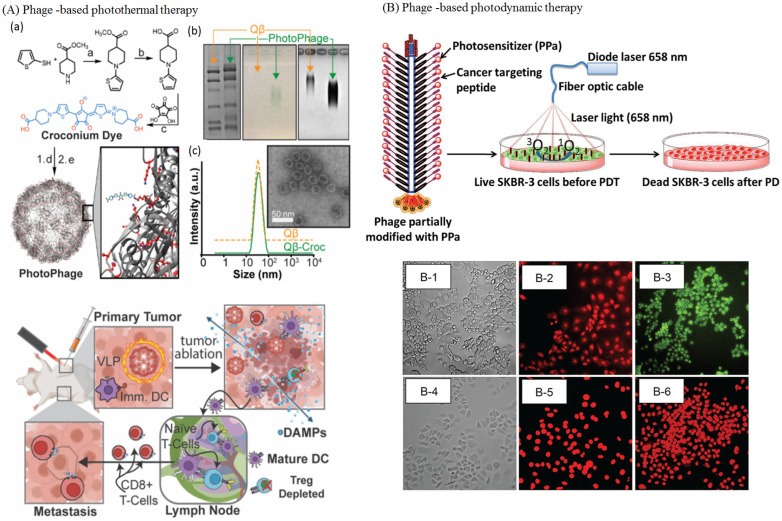
Phages in PTT and PDT. (**A**) Bacteriaophage Qβ-based PhotoPhage, for effective photo-immunotherapy. (a) Croconium dye synthesis. (b) Electrophoresis confirmed Croconium dye-Qβ bioconjugation. (c) Dynamic light scattering (DLS) and transmission electron microscopy (TEM) analysis of Qβ and PhotoPhage [232] (permission obtained for reuse of figure). (**B**) Targeted phage nanofibers with VSSTQDF and partial pyropheophorbide-a conjugation showed promising PDT effects on cancer cells [241] (citation credited and no permission needed). Targeted PDT validated by microscopy, fluorescence imaging of MCF-7 (B-1, B-2, and B-3) and SKBR-3 cells (B-4, B-5, and B-6).

#### 4.5.4. Phages in Cancer Immunotherapy

The body’s innate immune system includes cells and molecules that differentiate between self and non-self. They sense the environment by specific sensors called pattern recognition receptors (PRRs). There are numerous PRRs that are specific to mammalian pathogens. PRRs can be categorized into various types, such as toll-like receptors (TLRs), cytosolic DNA sensors (CDSs), nucleotide-binding oligomerization domain (NOD)-like receptors (NLRs), retinoic acid-inducible gene I (RIG-I) like receptors (RLRs), C-type lectin domain (CTLD) proteins, and absent in melanoma (AIM)-like receptors (ALRs) [242]. Recent studies have demonstrated that phages primarily interact with TLRs, which are expressed on the cell surface, in endocytic compartments, and in the cytoplasm [242,243]. It has been observed that phagocytes recognize phages through surface TLRs, while their encapsulated nucleic acids are recognized by cytoplasmic TLRs, leading to the secretion of cytokines and the induction of anti-tumor immunity [244]. PRRs like TLR3, TLR9, and possibly TLR7 may detect these phages intracellularly, while PRRs like TLR2 may sense them extracellularly. The naturally antigenic coat proteins of the phage head and the CpG islands in the phage genome induce an innate immune response [245]. Pathways involving sensing of single-stranded DNA (ssDNA) and double-stranded DNA (dsDNA) and the induction of IFN responses are most commonly implicated (Figure 6A,B) [246]. Antigen-presenting cells (APCs) can detect phages via intracellular (TLR3, TLR7, TLR9) and extracellular (TLR2, TLR4) PRRs, activating T/B cells for inflammatory responses and adaptive immunity, leading to anticancer antibody production through MHC-mediated peptide presentation and cytokine release. TLR9 recognizes unmethylated CpG motifs abundant in the DNA of phages as well as the bacteria that produce them. An oral cocktail of *E. coli* tailed phages led to significantly increased IFN-γ-producing CD4+ T cells, driven by DC sensing of phage DNA through TLR9 [244,247]. Another study made a discovery that a specific group of lytic phages can induce the production of Type I Interferon in a manner dependent on TLR9. When innate immune cells identify phages, they release cytokines that broadly activate T and B cells, including interferon (IFN)-γ, interleukin (IL)-6, IL-10, and IL-12. Activated T cells have the ability to produce IFN-γ and other cytokines, which can then mediate inflammatory responses. In addition to being capable of processing phage peptides, antigen-presenting cells can present them to T cells via MHC I and MHC II. This allows for antigen-specific antiphage adaptive immune responses. B cells can then be stimulated by antigen-specific T cells to generate antiphage antibodies [248]. These properties present a range of opportunities regarding the utilization of phages in cancer immunotherapy. This includes the potential to reprogram the tumor microenvironment (TME) due to their inherent immune-stimulatory properties. Additionally, phages can serve as effective carriers for delivering immunotherapeutic agents and play a crucial role in vaccine design.

TME exhibits a significant degree of immunosuppression and inclines towards facilitating the evasion of tumor cells from immune surveillance. This is achieved through the inhibition of antitumor T-cell generation, activation, and efficacy. As certain phages exhibit a high degree of immunogenicity, the administration of said particles into the tumor microenvironment has the potential to induce activation of the innate immune system, thereby initiating the progression towards adaptive antitumor immunity [250]. For example, it is demonstrated that inhalation of Ff phages lacking any proteins or peptides possesses the ability to impede the growth of glioblastoma tumors in a mouse model. By utilizing the intranasal route, a non-invasive method of administering therapeutics directly to the central nervous system, it has been demonstrated that these phages rapidly accumulate within the brains of mice and have the potential to mitigate the progression of orthotopic glioblastoma [251]. Sweere et al. reported that Pf, a filamentous *P. aeruginosa*-infecting phage, induced IFN-β production in DCs in a TLR3- and TRIF-dependent manner. Phage-derived RNA production within eukaryotic cells was demonstrated as the stimulus for this RNA-sensing receptor, although it is not yet understood how Pf is able to initiate transcription in a mammalian cell [252]. The TME is crucial for cancer progression and metastasis, regulating the differentiation of precursor monocytes into anti-tumor (M1) and pro-tumor polarized macrophages (M2). *E. coli* phage lysate can modify the TME, transforming tumor-associated M2 macrophages into anti-tumor M1 macrophages. Bacterial phage lysates (BPLs) and phage/BPL-coated proteins can also modify the TME, eliciting robust anti-tumor responses and facilitating the conversion of M2-polarized TAMs to a more M1-polarized environment [253].

Phages can be used as cancer vaccines due to their intrinsic immunogenicity, which triggers cellular and molecular reactions. As external antigens, phage particles activate the innate immune system and stimulate adaptive immunity in humans. This allows them to be used as a vehicle for carrier antigens, providing specific benefits in triggering cellular reactions [254]. Phage-displayed peptides, when processed, bind to the histocompatibility complex (MHC), inducing CD4+ and CD8+ T lymphocytes, thereby triggering robust cytotoxic reactions, crucial for anticancer vaccines [255,256]. Tumor-associated antigen (TAA) holds great potential as a target for cancer treatment. Methods to induce the production of antibodies against TAAs involve the administration of either full-length TAAs, their antigenic fragments, or TAA mimotopes, such as anti-idiotype antibodies or peptides that are recognized by the anti-TAA antibody. Extensive literature exists on the use of phage display panning techniques, utilizing either monoclonal or polyclonal antibodies, to identify TAA mimotopes. Firstly, conjugating a peptide with a phage particle elicits a superior response compared to conjugation with another carrier. Secondly, modifying the phage coat protein to reduce its complexity and immunogenicity redirects the immune response towards the peptide itself. Recently, the discovery of an anti-CD40 designed ankyrin repeat protein (DARPin) in M13 phages offers a bifunctional M13 phage for neoantigen delivery. M13CD40-based neoantigen vaccines show improved antigen retention and antigen accumulation in lymph nodes (LNs) compared to nontargeting phage vaccines. These vaccines also benefit from additional CD40 stimulation, enhancing antigen-specific immune responses and tumor control [198].

The capsids of phages are composed of multiple copies of their capsid proteins and serve as multivalent, repetitive scaffolds, thereby enhancing the multivalent presentation of antigens. These characteristics render phages as optimal platforms for the delivery and presentation of TAA and TAA mimotopes. Consequently, phage vaccines typically elicit a more robust immune response compared to soluble forms of antigens. Simultaneously, phages possess adjuvant properties, and numerous epitope display platforms do not necessitate the use of additional adjuvants to induce potent immunity. Covalent conjugation is the predominant approach employed for the attachment of tumor-associated antigens (TAAs) to phages. Phage display techniques have been increasingly investigated for vaccine design and delivery strategies in recent years. For example, the examination of phages in relation to the delivery of human epidermal growth factor receptor 2 (HER2) epitopes revealed that protective immunity and the potential capability of preventing relapse in HER2-positive breast cancer models [249]. Several HER2-based epitopes have been evaluated as phage-based cancer vaccines in pre-clinical research studies, such as AE37 (Ii-Key/HER-2/neu 776–790), H-2kd-restricted CTL, Δ16HER2 exposed, and peptide GP2 [257,258,259,260]. The study shows that anti-HER2 vaccination using the M13 bacteriophage platform induces a significant anti-HER2 antibody response and controls tumor growth in a breast cancer preclinical model, proving that anti-HER2 phage-based vaccines are a safe and successful immunotherapeutic strategy for HER2+ breast cancer patients (Figure 6B) [249].

Immunomodulator agents exhibit great promise in cancer immunotherapy due to their capacity to activate the immune system. Some immunomodulator agents with small-molecule properties have undergone clinical testing. However, their clinical application is impeded as a result of side effects and suboptimal pharmacokinetics. To surmount these limitations, they have been successfully loaded on phage nanoplatforms. For example, the linkage of 2-methoxyethoxy-8-oxo-9-(4-carboxybenzyl) adenine (1V209) as a TLR7 agonist on Qβ has reduced tumor growth in vivo and has extended the survival of mice in comparison to those treated with free 1V209 [261]. To this end, studies have demonstrated the incorporation of CpG oligodeoxynucleotide (ODN) into Qβ, which serves as a carrier [262,263]. CpG ODNs function as ligands for TLR9, and, upon activation, TLR9 possesses the ability to stimulate macrophages. The Qβ particles loaded with CpG were found to elicit more pronounced responses in cytotoxic T lymphocytes compared to CpG alone [262]. The study demonstrated that a 16.1-kD cytokine, GM-CSF, can be efficiently presented on M13 phage particles. The phage activates STAT5 signaling in murine macrophages and reduces tumor size by over 50% in a murine colorectal cancer model. Immunological profiling showed an increase in CD4+ lymphocytes in the GM-CSF treatment group [264]. Bacteriophages can deliver antitumor agents and advance vaccine development, but they often induce neutralizing antibodies. A genetically modified nonpathogenic bacterial strain is proposed to target tumors and release PD-L1-specific M13 bacteriophage. This phage-expressing strain, combined with a controlled immunotoxin release, has shown synergistic effects in inhibiting tumor growth in a colorectal cancer model. When combined with Folfox, the phage-expressing strain significantly extends survival. This strategy offers an effective and safe method for targeted therapeutic phage delivery to tumors [196].

#### 4.5.5. Phages in Combination Therapy

Combination strategies integrate therapies to overcome tumor heterogeneity. Phages, with their inherent immunomodulatory properties and ability to load therapeutic agents, offer new promise for combination therapies. For example, a new immunophotothermal agent, phage Qβ, uses chemically modified VLP for adjuvant photothermal ablation. This system converts croconium dyes to lysine residues, generates more heat, and is biodegradable. Its combination of thermal ablation and mild immunogenicity leads to effective tumor suppression, reduced lung metastasis, and increased survival time [232].

Combining GM-CSF phage therapy with radiation improved therapeutic potency, with a 100% survival rate and 25% complete remission rate (Figure 7) [264].

The M13 phage can remodel the tumor microenvironment, improving breast cancer treatment efficacy. The M13 gel, an engineered phage gel, can synthesize photothermal palladium nanoparticles (PdNPs) on pVIII capsid protein, forming M13@Pd Gel. This gel, loaded with NLG919, can downregulate the expression of the indoleamine 2,3-dioxygenase 1 enzyme. In vitro and in vivo studies show that the M13 gel acts as a self-immune adjuvant, effectively causing tumor cell death and down-regulating IDO1 expression [265].

#### 4.5.6. Phages as Bioimaging Agents

Postponed cancer diagnosis can increase mortality rates, making timely detection crucial for effective treatment [266]. Bio-imaging techniques can enable preclinical diagnosis, patient condition monitoring, and easy identification of pathological tissue during surgical procedures. Research on nanomaterials such as quantum dots, gold nanoparticles, silica nanoparticles, polymers, and VLPs has led to extensive advancements in bio-imaging technology [209]. A magnetic resonance imaging approach is employed that uses P22 phages. GdIII-chelating agents are affixed to either the inner or outer surface of P22 viral capsids. This system enables the non-invasive visualization of the intravascular system, as demonstrated in the magnetic resonance (MR) image [267]. Researchers have developed a new technique for labeling filamentous phage capsid proteins by converting N-terminal amines into ketone groups. This allows for the attachment of fluorophores and up to 3000 molecules of 2 kDa poly(ethylene glycol) (PEG2k) to each phage capsid without affecting antibody binding to EGFR and HER2. The modified phage is also useful for breast cancer cell characterization [268]. The study uses the T4 phage head as a scaffold for bioconjugating fluorescent dyes for cell imaging and flow cytometry applications. The large surface area of the T4 head allows for larger functional groups, such as fluorescent dyes. Cy3 and Alexa Fluor 546, which were chemically incorporated into tail-less T4 heads, resulted in fluorescent properties that were characterized. The dye-conjugated T4 nanoparticles showed up to 90% enhancement in fluorescence compared to free Cy3. The dye-conjugated nanoparticles are structurally stable and can be used as molecular probes for these applications [269]. Anti-EGFR antibodies were conjugated to MS2 capsids to create nanoparticles targeting breast cancer cells. These agents showed good stability and specific binding in in vitro experiments. They were injected into mice with tumor xenografts, and their localization was determined using PET/CT and scintillation counting. The capsids showed long circulation times and moderate tumor uptake, with 10–15% ID/g in the blood at 24 h [270]. Utilizing targeting peptides, AF680-labeled phage nanoparticles are employed in the imaging of ovarian cancer cell lines by means of fluorescent microscopy [271]. In order to visualize HER3-positive cancer through positron emission tomography (PET), the phage display technique was utilized to isolate an anti-HER3 antigen-binding fragment that serves as a near-infrared fluorescence imaging probe [272]. A study uses phage display to screen human lung adenocarcinoma-specific peptides for cancer diagnosis. The highest frequency peptide, Pep-1 (a specific peptide sequence (CAKATCPAC)), was identified for imaging probe capabilities. This peptide sequence is a promising diagnostic lead for rapid and accurate detection of lung adenocarcinoma, suggesting potential use for prognostic diagnosis after radiotherapy [273]. The M13 bacteriophage has become an appealing bionanomaterial due to its ability to manipulate its surface chemistry and its potential to self-assemble into complex structures. Techniques involving chemical modification have been utilized to incorporate a wider array of imaging agents on the M13 bacteriophage (Figure 8) [274].

#### 4.5.7. Phages in Theragnostics

Phages can merge imaging and therapeutic agents in a single platform. Their inherent therapeutic potential, targeting ability, and the capability to serve as carriers for therapeutic and imaging agents suggest their potential as theranostic agents [228]. The multifunctional phage M13, conjugated with chemotherapy, fluorophores, and targeting ligands, enables simultaneous imaging and drug delivery to prostate cancer cells [215]. A bioinspired phage nanosome coated with gold nanoparticles (ΦNSAu) has been found to enhance the optical properties of dyes, making them excellent imaging agents. The nanosomes, when combined with chemotherapeutic drug Mitoxantrone (ΦNSAuM), showed excellent photothermal transduction efficacy, exhibiting anti-cancer activity against 4T1 cell lines. The phage-based nanosomes also demonstrated potential as a photothermal agent, demonstrating their potential for anti-cancer theranostics [275]. Figure 9 shows HCC-specific MS2 VLPs containing cross-linked therapeutic and imaging agents [204].

## 5. Recent Developments in Phage Engineering and Therapy

The collection of research results underscores the remarkable potential of engineered phages in fighting antibiotic-resistant bacteria, cancer, and superbugs [276]. Biological and chemical methods have been created for engineering phages and generating therapeutic compounds. The most widely used technique is phage display. While traditional phage display can produce billions of peptides at once, it is restricted to using only canonical amino acids. Recently, noncanonical amino acids (ncAAs) with unique reactivities and chemistries into phage-displayed peptide libraries have been successfully expanded. The incorporation of ncAAs and specific motifs simplifies the conversion of peptide lead compounds into therapeutics by enhancing their stability and target binding [277]. In a study, Chen et al. demonstrated this novel approach’s effectiveness by identifying potent inhibitors for the ENL YEATS domain that plays a critical role in leukemogenesis. Their strategy involved genetically incorporating Nε-butyryl-l-lysine (BuK), known for its binding to ENL YEATS, into a phage display library for enriching the pool of potent inhibitors. This led to the creation of selective ENL YEATS inhibitors with a kD value of 2.0 nM and a selectivity 28 times higher for ENL YEATS than its close homologue AF9 YEATS. One such inhibitor, tENL-S1f, demonstrated robust cellular target engagement and on-target effects to inhibit leukemia cell growth and suppress the expression of ENL target genes. As a pioneering study, this work opens up extensive avenues for the development of potent and selective peptidyl inhibitors for a broad spectrum of epigenetic reader proteins [278].

Recently, for addressing the solubility and stability issues of organic NIR dyes for enhancing near-infrared (NIR) phototherapies, which are noninvasive and cost-effective for treating tumors and infections, IR780 and Indocyanine green (ICG) encapsulated within phage nanosomes were used. The results improved the dyes’ UV–vis absorbance and photothermal efficacy compared to liposomes. Experimental and molecular dynamics simulations confirmed better nanoscale structure, solubility, dynamics, and binding of these dyes to the phage capsid. These dye-loaded phage nanosomes, coencapsulated with mitoxantrone, showed enhanced anticancer activity and, when combined with amphotericin B, exhibited superior photothermal effects against fungal infections [279].

To develop phage therapy against a diverse range of clinically relevant Escherichia coli, a library of 162 wild-type (WT) phages was screened, identifying eight phages with broad coverage of *E. coli*, complementary binding to bacterial surface receptors, and the capability to stably carry inserted cargo. Selected phages were engineered with tail fibers and CRISPR–Cas machinery to specifically target *E. coli*. It showed that engineered phages target bacteria in biofilms, reduce the emergence of phage-tolerant *E. coli*, and out-compete their ancestral WT phages in coculture experiments. A combination of the four most complementary bacteriophages, called SNIPR001, is well tolerated in both mouse models and minipigs and reduces *E. coli* load in the mouse gut better than its constituent components separately. SNIPR001 is in clinical development to selectively kill *E. coli*, which may cause fatal infections in hematological cancer patients [280].

Customizable and number-tunable enzyme delivery nanocarriers will be useful in tumor therapy. A phage vehicle, T4-Lox-DNA-Fe (TLDF), which adeptly modulates enzyme numbers using phage display technology to remodel the tumor microenvironment (TME), is presented [281]. Regarding the demand for lactic acid in tumors, each phage is engineered to display 720 lactate oxidase (Lox), contributing to the depletion of lactic acid to restructure the tumor’s energy metabolism. The phage vehicle incorporated dextran iron (Fe) with Fenton reaction capabilities. H_2_O_2_ is generated through the Lox catalytic reaction, amplifying the H_2_O_2_ supply for dextran iron-based chemodynamic therapy (CDT). Drawing inspiration from the erythropoietin (EPO) biosynthetic process, an EPO enhancer has been constructed to impart the EPO-Keap1 plasmid (DNA) with tumor hypoxia-activated functionality, disrupting the redox homeostasis of the TME. Lox consumes local oxygen, and the positive feedback between the Lox and the plasmid promotes the expression of kelch ECH Associated Protein 1 (Keap1). Consequently, the downregulation of the antioxidant transcription factor Nrf2, in synergy with CDT, amplifies the oxidative killing effect, leading to tumor suppression of up to 78%. This study seamlessly integrates adaptable T4 phage vehicles with bio-intelligent plasmids, presenting a promising approach for tumor therapy [281].

PDT targets cancer cells by converting oxygen into reactive oxygen species (ROS), but its effectiveness is reduced in hypoxic (low oxygen) tumor environments. To address this, researchers propose engineering filamentous fd phage, a virus specific to bacteria and safe for humans, into a nanozyme-nucleating, photosensitizer-loaded, tumor-homing nanofiber to boost ROS production in hypoxic tumors. Genetically modified fd phage displays platinum (Pt)-binding peptides on its sidewall and tumor-homing peptides on its tip. These Pt-binding peptides facilitate the formation and alignment of Pt nanozymes (PtNEs) on the phage’s sidewall. The resulting PtNE-coated, tumor-homing phage significantly enhances the sustained catalytic conversion of hydrogen peroxide into oxygen in hypoxic tumors, improving ROS production for PDT compared to non-phage-templated PtNEs. Density functional theory (DFT) calculations confirm the catalytic mechanism of the phage-templated PtNEs. When injected intravenously into mice with breast tumors, these PtNE-coated, indocyanine green (ICG)-loaded phages target the tumors and effectively inhibit tumor growth through PtNE-enhanced PDT. Additionally, the nanofibers serve as tumor-homing imaging probes due to the fluorescence of ICG. This study demonstrates that engineered filamentous phage can act as cancer-targeting nanozymes with enhanced catalytic performance for effective targeted PDT [239]. It was demonstrated that encapsulating the photosensitizer zinc naphthalocyanine within cationic liposomes and integrating them with anionic M13 phages into a nanoweb structure significantly improves photosensitizer delivery to cancer cells, potentially enhancing PDT effectiveness [282].

PDT represents an emerging strategy to treat various malignancies, including colorectal cancer (CC), the third most common cancer type. This work presents an engineered M13 phage retargeted towards CC cells through pentavalent display of a disulfide-constrained peptide nonamer. The M13CC nanovector was conjugated with the photosensitizer Rose Bengal (RB), and the photodynamic anticancer effects of the resulting M13CC-RB bioconjugate were investigated on CC cells. This showed that, upon irradiation, M13CC-RB is able to impair CC cell viability, and that this effect depends on (i) photosensitizer concentration and (ii) targeting efficiency towards CC cell lines, proving the specificity of the vector compared to unmodified M13 phage. It was also demonstrated that M13CC-RB enhances the generation and intracellular accumulation of reactive oxygen species (ROS), triggering CC cell death. To further investigate the anticancer potential of M13CC-RB, this study performed PDT experiments on 3D CC spheroids, proving, for the first time, the ability of engineered M13 phage conjugates to deeply penetrate multicellular spheroids. Moreover, significant photodynamic effects, including spheroid disruption and cytotoxicity, were readily triggered at picomolar concentrations of the phage vector. Taken together, these results promote engineered M13 phages as promising nanovector platforms for targeted photosensitization, paving the way to novel adjuvant approaches to fight CC malignancies [192].

For many years, bacteriophages were believed to be entirely neutral towards eukaryotic cells, including those of animals and humans, due to their specificity for bacterial hosts. However, recent studies reveal that phages can indeed interact with eukaryotes in both direct and indirect ways—directly influencing cells, tissues, and organs, or indirectly altering the microbiome [283]. These interactions can significantly impact various systems, such as the immune, respiratory, central nervous, gastrointestinal, urinary, and reproductive systems, and even play a role in cancer dynamics. Their effects, which can be either beneficial or harmful, highlight the importance of thoroughly understanding the mechanisms behind phage-eukaryote interactions. Such knowledge is essential for effectively harnessing bacteriophages in medicine, particularly phage therapy and biotechnology, leveraging their potential to combat bacteria and modulate eukaryotic functions wisely.

## 6. Conclusions

Efficient delivery of therapeutic agents to target sites remains a critical challenge in disease management. Nanotechnology, particularly the use of bacteriophages, offers promising solutions. Over a thousand different types of phages infect bacteria. They thus constitute the most predominant form of biological particles in the world. Bacteriophages exhibit beneficial properties, such as biocompatibility, high drug-loading capacity, immune evasion, and ease of modification, making them effective nanocarriers. They can be customized for diverse applications, including capsid-associated display, chemical conjugation, polymer coating, and encapsulation. These capabilities enable targeted delivery in cancer therapy, bacterial/viral infections, and in gene therapy, addressing conditions that are difficult to treat through conventional methods. The multiple uses of phage in human health are still being elucidated. This review has discussed various applications of phage therapy, such as the fact that phages loaded with a therapeutic agent payload can bind to, penetrate, and inhibit the growth of target cells. However, even as phage therapy is propitious in the treatment of several diseases, it is presently more established and has achieved better success in the therapy of bacterial infections, in particular those showing resistance to antibiotics rather than functioning as direct therapeutics of cancer, viral diseases and autoimmune diseases as seen by the predominant wealth of literature accumulated addressing the phage treatment of bacterial diseases. Phage-based cancer therapy is still in the initial stages of research, and further studies are required to fully comprehend its potential and drawbacks. Engineered phages are currently being generated as potential vaccines against cancer with the aim of obviating cancer or promoting treatment outcomes. Despite regulatory and safety concerns, ongoing research efforts continue to refine bacteriophage-mediated systems, paving the way for innovative therapeutic approaches that may overcome these challenges and unlock broader clinical applications.

## Figures and Tables

**Figure 1 bioengineering-12-00469-f001:**
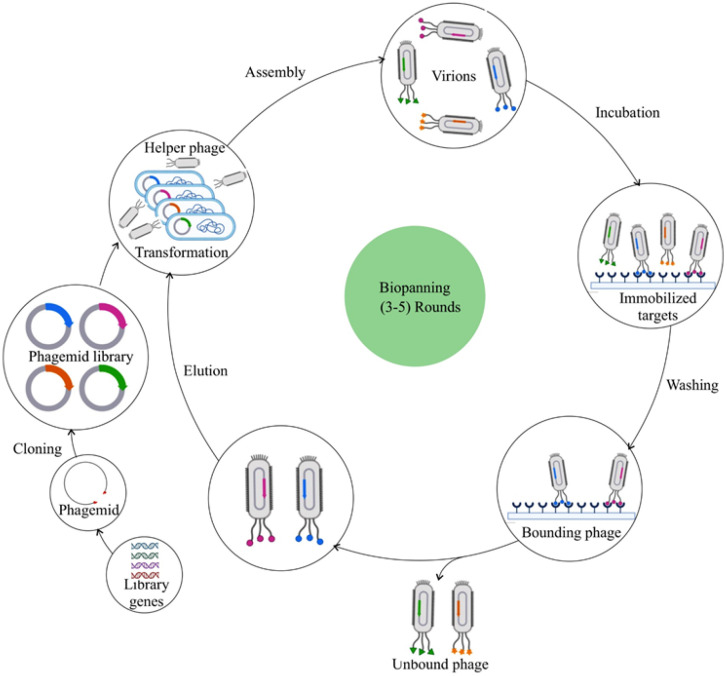
The principle of bacteriophage display using a phagemid vector involves the following steps: Genes encoding millions of variant libraries are inserted into a phagemid vector. Large phage libraries are generated by transforming bacteria with these phagemids, followed by phage rescue using helper phages. Phages displaying specific-binding ligands to immobilized targets are then identified through iterative biopanning rounds, consisting of binding, washing, elution, infection, and amplification (own figure). The red, green and blue colors in the phagemid libraries represent genes encoding different antibodies. These colors on the phage virions represent different antibodies.

**Figure 2 bioengineering-12-00469-f002:**
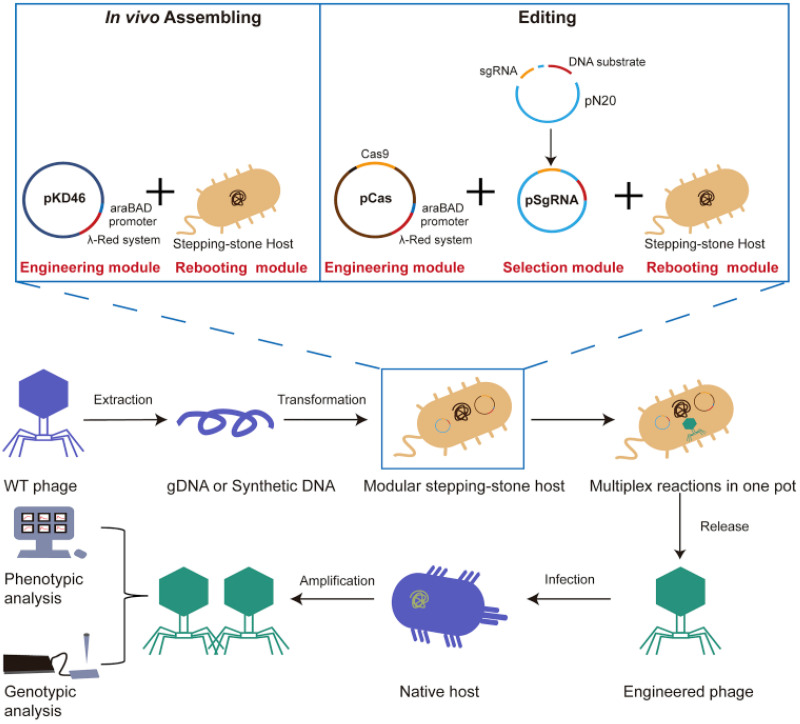
SHAPE has two functions: in vivo assembly and DNA editing. SHAPE stands for stepping-stone host-assisted phage engineering. This cell-free system includes the engineering module, prior to transformation into the stepping-stone host. For in vivo assembly, the plasmid pKD46 is transformed into the stepping-stone host. The synthetic DNA fragments are transformed into the stepping-stone host harboring the plasmid pKD46, and de novo synthetic phages are produced by the stepping-stone host and amplified on a lawn of a natural host. For DNA editing, sgRNA and DNA substrates are cloned in the pN20 vector. The resulting pSgRNA is co-transformed with pCas into the stepping-stone host. The phage genome is transformed into the stepping-stone host harboring the two plasmids, and engineered phages are produced by the stepping-stone host and amplified on a lawn of a natural host [83] (citation credited and no permission needed for reuse of figure).

**Figure 3 bioengineering-12-00469-f003:**
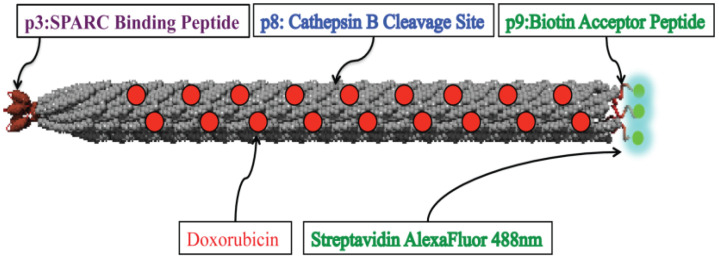
Phages in targeted drug and gene delivery. M13 phage modification provides a nanosized delivery system that is capable of loading antitumor drugs such as doxorubicin (DOX). The red dots along the phage coat represent DOX attached to the gene-8 major coat protein (p8). The gene-3 minor coat protein (P3) displays a peptide with affinity for secreted protein acidic and rich in cysteine (SPARC), and p9 coat protein (p9) can be enzymatically biotinylated and loaded with streptavidin functionalized fluorophores [215] (citation credited and no permission needed for reuse of figure).

**Figure 4 bioengineering-12-00469-f004:**
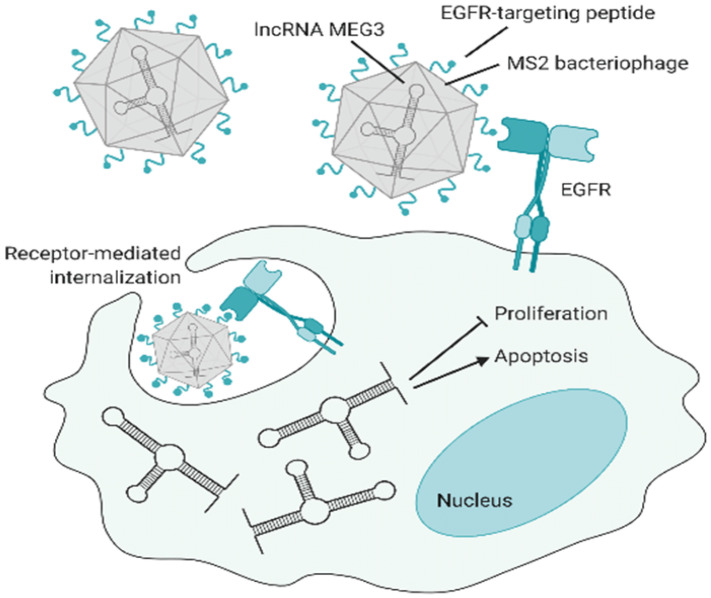
The induction of apoptosis and inhibition of cancer cell proliferation is achieved through the reintroduction of maternally expressed gene 3 long non-coding RNA (MEG3 lncRNA) in cancer cells. This reintroduction is facilitated by an epidermal growth factor receptor (EGFR)-targeted MS2 phage, which binds to the EGFR receptor on the cancer cells. Upon binding, the phage is internalized via endocytosis and subsequently releases MEG3 [228] (citation credited and no permission needed).

**Figure 6 bioengineering-12-00469-f006:**
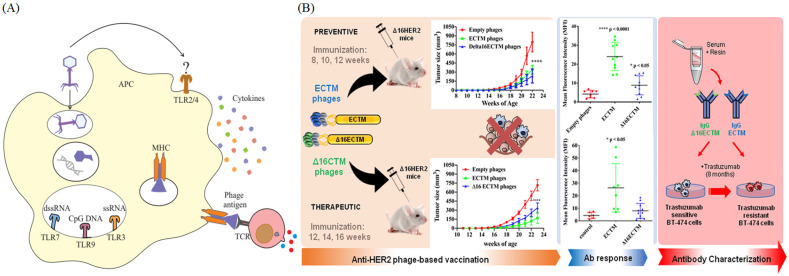
Pathways involving sensing of phages and the induction of immune responses. (**A**) Antigen-presenting cells (APCs) recognize phages via intracellular (TLR3/7/9) and extracellular (TLR2/4) PRRs, triggering cytokine release. Phages are processed, presented on MHC molecules, activating T/B cells for anticancer immunity. (**B**) Vaccination with M13 bacteriophages carrying HER2 or D16HER2 delays mammary tumor onset and reduces growth in D16HER2 transgenic mice. Data represent Mean fluorescence intensity (MFI) ± SD; Student’s *t*-test (**** *p* < 0.0001; * *p* < 0.05) [249] (citation credited and no permission needed).

**Figure 7 bioengineering-12-00469-f007:**
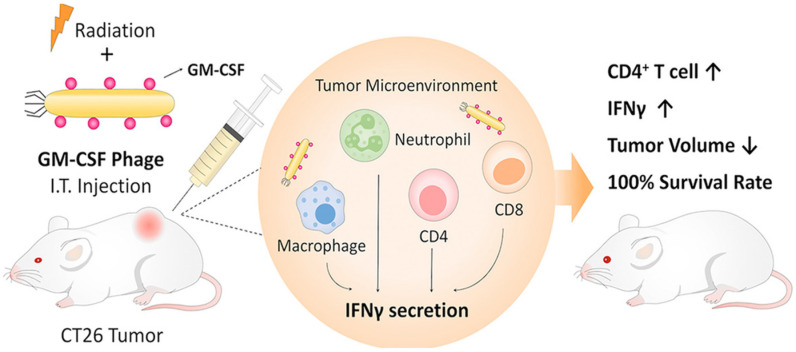
Phages in combination and theranostic therapy.The study developed a new therapeutic platform using filamentous phages, utilizing pVIII for antitumor cytokine display. The GM-CSF phage was efficiently presented on M13 phage particles, activating STAT5 signaling in murine macrophages. The phage reduced tumor size by over 50% in a murine colorectal cancer model. Combining therapy with radiation improved therapeutic potency, with a 100% survival rate and 25% complete remission rate [264] (permission obtained for reuse of figure). ↑ represents an increase and ↓ indicates a decrease.

**Figure 8 bioengineering-12-00469-f008:**
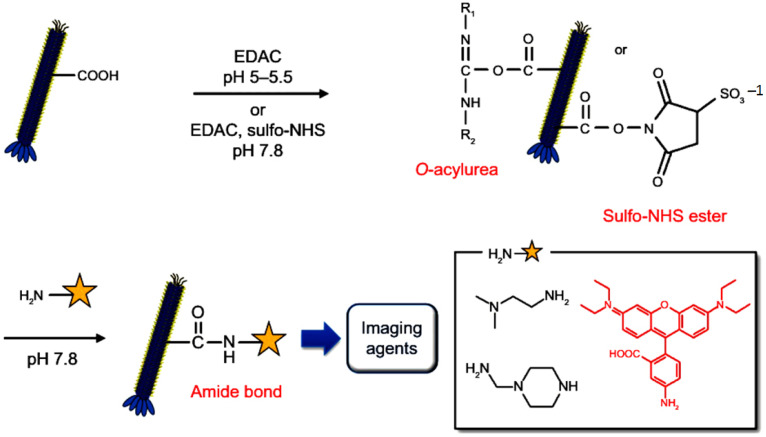
Chemical modification of phages for bioimaging. Fluorescent moieties could be attached to phage surface via reactive groups to create imaging agents [274] (citation credited and no permission needed).

**Figure 9 bioengineering-12-00469-f009:**
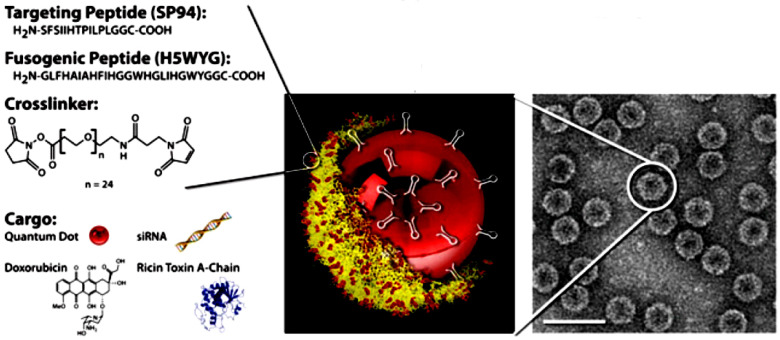
The process involves synthesizing HCC-specific MS2 VLPs, which contain therapeutic and imaging agents, by conjugating nanoparticles, protein toxins, and drugs using a cross-linker. Transmission electron microscopy (TEM) image is depicted loaded capsids(scale bar = 50 nm) [204] (citation credited and no permission needed).

**Table 2 bioengineering-12-00469-t002:** Examples of successful studies using engineered phage to treat viral infections.

Engineered Phage	Findings	In Vitro/In Vivo Models	Reference
HBcAg core antigen was fused to the M13 phage pIII	Immunization with these recombinant phages showed potent immunogenicity	Mice models	[147]
HSV-1 glycoprotein D expression cassette was inserted into M13 phage genome	Recombinant phage upon expression functioned as a potent vaccine with high capability to induce cell mediated immunity and neutralizing antiviral antibodies	Mice models	[148]
SARS-CoV-2 spike (S) protein and the CAKSMGDIVC peptide were respectively displayed on the phage M13 pVIII and pIII	Elicited robust specific and systemic immune reactions with no adverse effects	Mice models	[149]
SARS-CoV-2 pneumonia	Infection successfully neutralized by phage therapy at a low dosage of 2 mg/kg in mice while in the hamster model, phage administration proved highly therapeutic and prophylactic	Mouse and hamster models	[150]
HIV antigens displayed on the T4 surface as many copies	Robust and broadly neutralizing antibodies and cell-mediated T-cell responses were elicited to HIV antigens in the absence of any external adjuvants	Mice models	[151]

**Table 3 bioengineering-12-00469-t003:** Some examples of the employment of phage therapy in treating veterinary diseases.

Engineered Phage	Findings	In Vitro/In Vivo Models	Reference
Phage cocktail to treat mastitis induced by *S. aureus*	Treatment with phage cocktail led to the highest intramammary phage titer when compared to other cohorts and possessed efficiency comparable to that induced by the antibiotic, ceftiofur sodium	Murine models	[129]
The novel peptidase derived from phages, CHAPK against Staphylococci involved in the formation of biofilms	Acted as an efficient biocidal agent enabling rapid disruption of bacterial biofilms suggesting that it can be incorporated in the teat-dip solution to preclude S. aureus colonization upon the udder skin surface of bovines	In vitro studies	[153]
Phage particles against *S. intermedius* causing pyoderma skin infections and collected from wounds and inner hearing channel of animals (dogs and horses)	Cutaneous permeation of phage particles conveyed in a hydroxyethylcellulose (HEC) gel and integrating ionic liquid that acted as a permeation enhancer; the ionic liquid highly enhanced transdermal permeation of the bacteriophage particles, with associated high potential of the HEC gel formulation in the antimicrobial treatment of animal skin infections	In vitro assays	[154]
Phage PIZ SAE-01E2 against *Salmonella enterica* subsp. enterica serovar Abortusequi infections causing abortion in mares and donkeys	Prophylactic and therapeutic effects observed wherein a single intraperitoneal injection of PIZ SAE-01E2 before or after bacterial challenge provided effective protection against abortions in all pregnant mice	Pregnant murine model of abortion	[155]
Phages P2S2 and P5U5 against multidrug-resistant pathogenic strains of *P. aeruginosa* sourced from canine skin diseases wherein *P. aeruginosa* is responsible for otitis externa in dogs, in addition to wound infection, chronic deep pyoderma, and ocular infections including ulcerative keratitis	Potent lytic activity against a wide range of *P. aeruginosa* strains obtained from canine ocular infections (80–100% lysis,); preparation containing both phages showed a notable inhibition of bacterial growth at all the MOIs tested	In vitro studies	[156,157]
Bacteriophages against uropathogenic multidrug-resistant *E. coli* strains in dogs and cats	Over 90% of the ten bacteriophages isolated were capable of lysing about 50% of the target *E. coli* sourced from feline and canine feces upon singular testing, and over 90% were able to lyse the target when administered as a cocktail	In vitro studies	[158]
A cocktail of φ26, φ27, and φ29 bacteriophages against Shiga-toxin-expressing *Escherichia coli* responsible for causing neonatal diarrhea	Suppositories containing a cocktail of the three *E. coli* lytic phages and *Lactobacillus* spp. (a probiotic bacterium) showed both prophylactic and therapeutic effects without impacting endogenous microflora	In vivo testing in calves	[159]
A cocktail of lytic bacteriophages SPFM14 and SPFM10 against *S. typhimurium* challenge in pig gastrointestinal tracts	Upon prophylactic oral administration in feeds, demonstrated a significant decrease in the colonization of target bacteria	In vivo studies in pigs	[160]
Two new lysins sourced from lysogenic phages (phi5218 and phi7917) targeting *Streptococcus suis* multiple serotypes	Efficient lytic activity and therapeutic potential	In vitro and in vivo studies in mouse and piglets	[161,162,163]
Virulent phage CP220 administered to broiler chicken infected with *Campylobacter coli* and *Campylobacter jejuni*	Notably lower target bacteria count within the intestines	In vivo studies in broiler chicken	[164]
A cocktail of 8 bacteriophages against avian pathogenic *E. coli* challenge	90% protection from death in comparison to control eggs, which showed 100% mortality	In ovo inoculation into embryonated eggs	[165]

**Table 4 bioengineering-12-00469-t004:** Some examples of the employment of phage therapy in treating cancers.

Therapies	Phage Type	Strategy	Therapeutic Agent	Cancer	Reference
Chemo-therapy	Phage A54	conjugation	doxorubicin (DOX)	hepatocarcinoma cells	[190]
	Phage fUSE5-ZZ	conjugation	hygromycin and doxoru-bicin	breast carcinoma cell lines	[191]
Photodynamic therapy (PDT)	M13 phage	conjugation	Rose Bengal (RB)	colorectal cancer (CC)	[192]
Gene therapy	M13	Display	TRAIL gene	Hepatocellular carcinoma (HCC),	[193]
	M13	Display	TRAIL gene	Chondrosarcoma (CS)	[194]
	Lamda (λ ZAP-CMV)	Display	Apoptin	breast carcinoma cell lines	[195]
Immuno-therapy	M13	Display	Programmed death ligand 1 (PD-L1)	colorectal cancer	[196]
	lambda	Display	Trastuzumab	Breast cancer	[197]
	M13	Display	anti-CD40 designed ankyrin repeat protein (DARPin)	Breast cancer	[198]
	fd	Display and biopan-ning	PD-L1-binding peptide (HH) and a melanoma-targeting peptide (IP)	Melanoma	[199]
	M13	biopanning	proapoptotic peptide, D(KLAKLAK)	breast cancer	[200]
Combination therapy	M13	Display	CD40+ 10-hydroxycamptothecin (HCPT), +PD-1 blockade		[201]

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
