# Peer review of "Bacteriophages as Targeted Therapeutic Vehicles: Challenges and Opportunities"

_bioengineering, 2025, doi:10.3390/bioengineering12050469_

Round 1
Reviewer 1 Report
Comments and Suggestions for Authors
Bacteriophages are highly specific to the bacterial cell that they infect and recognize many bacterial proteins that can serve for targeted therapy. They are easy to produce and deliver to cells. Therefore, bacteriophages are considered as attractive candidates for new DNA delivery. The review is devoted to the latest achievements in the field of biotechnology using bacteriophages.
The review discusses studies that provide evidence of the best use of a cocktail from phages, which increases the spectrum of bacterial targets. The review shows the results of studies that convincingly indicate that the phage -therapy serves as a safe alternative to the treatment of inflammatory infections of the urinary tract, the gastrointestinal tract and other organ systems with antibiotics or can be used in combination with antibiotics. The review provides data that the phage-based therapy for human burns led to a complete recovery in 86% of the treating patients, the rest had a noticeable improvement. Infections in diabetic legs were successfully treated with phage-based therapy without recurring infection, tissue disintegration or side effects. Success in therapy with phages are shown for many bactrial infections in both humans and model mice. The article has a review of the research on the influence of phage-based therapy on treatment of Oral Infections, Infections of the Gastrointestinal System, Infections of the Respiratory System, Infections of the Urinary Tract, Infection of the Eyes, Infections of the Ears, Nasal Infections, Complications Due to Bacteremia/Sepsis, SARS-CoV-2 Pneumonia, Liver Disease, Orthopaedic Infections and Bovine Mastitis.
One of the aspects discussed in the review is to study the use of phages in the treatment of bacterial infections resistant to antibiotics. Due to their high specificity to bacterial owners, phages can be aimed at numerous drug resistance (MDR) bacteria without the influence of a useful microbiota. A review of research is given regarding the use of phages in antiviral therapy.
Production of mAbs using Phage Display Technology and Phage Display Technology in diagnostics and therapy of neurodegenerative diseases, such as Alzheimer’s disease, also discussed in the review. Researchers have developed a SpyPhage system, a method for engineering phages with a SpyTag moiety, allowing for rapid surface modification with therapeutic proteins fused with SpyCatcher. This can revolutionize the phage of oncological diseases. Current trends in phage-based tumor treatments include targeted delivery of therapeutic agents, tumor-targeted immunotherapies, and combinational therapies. Fagas are also used as Bioimaging Agents for the timely detection of cancer cells, which is crucial for effective treatment.
Moreover, the advantages and disadvantages of the use of phage-based therapy are also discussed along with regulatory issues concerning their use in disease prophylaxis and therapeutics. The authors indicate that the advantages of therapy with phages include the prevalence of phages in nature, the ease of genetic manipulations with phages, their stability at different temperatures and the possibility of delivery of the phage to the place of tumor for targeted struggle. From a review of publications, the authors conclude that the disadvantages include the need for a large number of phage particles, the immunity of bacterial cells, the release of bacteriotoxins during cell lysis, the need to combine high phage virulence for bacterial targets and lowered potential for transfer of bacterial genes between different bacterial targets. A major limitation of phage therapy is the fact that phages are highly specific with respect to their hosts and thus cannot infect all of the sub-strains of a particular pathogenic strain. To overcome this, the latest genetic technology, such as a CRISPR-Cas9 system, is used to modify the phages. The review also discusses the restrictions in the regulation of the use of phage-based therapy, as well as differences in regulation in different countries.
The list of references contains 259 sources. Most of these articles have been published in the past five years. This is a booming field of biology and medicine and this review gives understanding of ​​the state of research.
Author Response
We welcome the reviewer's positive comments about our manuscript. We appreciate the efforts that the reviewer has put into the detailed report on the extensive coverage of literature presented in our paper.
Reviewer 2 Report
Comments and Suggestions for Authors
Manuscript described the delivery of therapeutics using bacteriophage vectors. Review has a potential, but it needs some of work and re-organization.
My comments are reported in the following:
- The title indicates “Bacteriophage Vectors”, but the article itself is devoted to phage therapy and its individual aspects. In fact, there is no description of the study of bacterial vectors. Please correct title or body manuscript.
- There are a huge number of reviews on phage therapy. The Authors should make their review stand out in some way.
- The description and material presentation are very superficial for review article.
- Please correct the article structure.
- The title, purpose, and conclusion should be derived from each other.
- The abstract should have been rewritten. In the abstract, the authors should mention the main most interesting points of the review for the reader and the results obtained. Instead, the authors try to formulate the purpose of the paper.
-In Advantages of Phage Therapy Section please add information about phage cocktails and synergism with antibiotics.
-The disadvantages of phage therapy section is not clear. There is no clear description of phage therapy disadvantages. Please make the structure in this section. For ex., Authors wrote “Nevertheless, since phages can often be used along with other phages (phage cocktails) and antibacterial agents, the phage products’ lytic spectrum maybe wider than the activity spectrum of in-dividual phage strains [17, 18].” – it is advantages or disadvantages?
- More figures will help to understand the studies using 'phage display' (The Technology of Phage Display and Phage Display Biopanning Strategy sections).
- Phage display can be used to obtain not only peptides, but for antibodies. Please add clear information in Phage Display Biopanning Strategy Section.
- Use of Phage Cocktails in Phage Therapy Section is not informative and not provided the prospects of using phage cocktails. - Just general words.
- There are a lot of good reviews on phage display but Authors not used its. Please correct it.
- Phage therapy is applicable not only in medicine, but also in veterinary medicine. Please add information.
- Please add some information about the features of application and development of therapy based on bacteriophages taking into account the impact of individual physiological and pathophysiological factors of the body.
- Authors wrote “Over a thousand different types of phages infect bacteria. They thus constitute the most predominant form of biological particles in the world..….” but not informaRegardless, the multiple uses of phage in human health are still being elucidated. This review has discussed various applications of phage therapy, such as the fact that phages loaded with a therapeutic agent payload can bind to, penetrate and inhibit the growth of target cells.”
- In vitro, Enterococcus faecalis, Mycobacterium tuberculosis etc. should be in italic.
- H2O2 - should be H2O2.
Author Response
Reviewer #2
Manuscript described the delivery of therapeutics using bacteriophage vectors. Review has a potential, but it needs some of work and re-organization.
My comments are reported in the following:
- The title indicates “Bacteriophage Vectors”, but the article itself is devoted to phage therapy and its individual aspects. In fact, there is no description of the study of bacterial vectors. Please correct title or body manuscript.
Response: We changed title to Bacteriophages as Targeted Therapeutic Vehicles: Challenges and Opportunities"
- There are a huge number of reviews on phage therapy. The Authors should make their review stand out in some way.
Response: We change organization
- The description and material presentation are very superficial for review article.
Response: we used many articles for presenting the depth, context, and critical analysis of the subject.
- Please correct the article structure.
Response: We change structure and organization
- The title, purpose, and conclusion should be derived from each other.
Response: We changed abstract and introduction.
- The abstract should have been rewritten. In the abstract, the authors should mention the main most interesting points of the review for the reader and the results obtained. Instead, the authors try to formulate the purpose of the paper.
Response: We change abstract
-In Advantages of Phage Therapy Section please add information about phage cocktails and synergism with antibiotics.
Response: We added a paragraph about comment.
-The disadvantages of phage therapy section is not clear. There is no clear description of phage therapy disadvantages. Please make the structure in this section. For ex., Authors wrote “Nevertheless, since phages can often be used along with other phages (phage cocktails) and antibacterial agents, the phage products’ lytic spectrum maybe wider than the activity spectrum of in-dividual phage strains [17, 18].” – it is advantages or disadvantages?
Response: We added a paragraph about comment.
- More figures will help to understand the studies using 'phage display' (The Technology of Phage Display and Phage Display Biopanning Strategy sections).
Response: We added an image about comment.
- Phage display can be used to obtain not only peptides, but for antibodies. Please add clear information in Phage Display Biopanning Strategy Section.
Response: We added some studies about comment.
- Use of Phage Cocktails in Phage Therapy Section is not informative and not provided the prospects of using phage cocktails. - Just general words.
Response: We added some studies about comment
- There are a lot of good reviews on phage display but Authors not used its. Please correct it.
Reviewer comment:
- Phage therapy is applicable not only in medicine, but also in veterinary medicine. Please add information.
Author rebuttal:
A table has been formulated to cover the successful use of phage therapy in treating veterinary diseases
- Please add some information about the features of application and development of therapy based on bacteriophages taking into account the impact of individual physiological and pathophysiological factors of the body.
Response: We added some information about comment in Recent developments in phage engineering and therapy.
- Authors wrote “Over a thousand different types of phages infect bacteria. They thus constitute the most predominant form of biological particles in the world..….” but not informaRegardless, the multiple uses of phage in human health are still being elucidated. This review has discussed various applications of phage therapy, such as the fact that phages loaded with a therapeutic agent payload can bind to, penetrate and inhibit the growth of target cells.”
Response: We changed conclusion
- In vitro, Enterococcus faecalis, Mycobacterium tuberculosis etc. should be in italic.
Response: We fixed them
- H2O2 - should be H2O2.
Response: We fixed them

Reviewer 3 Report
Comments and Suggestions for Authors
The review compiles a substantial amount of information but lacks a coherent narrative structure, making it difficult to read. The themes are disjointed, and locating specific information within the review is challenging.
In the introduction, the authors outline 10 objectives:
«The current review addresses the application of phages:
- in therapy of infectious diseases,
- neurodegenerative disorders,
- cancer
- and theragnostics
- with an insightful look into the phage display technology
- and biopanning strategy to generate engineered phages.
- The advantages
- and disadvantages of the use of phage-based therapy are also discussed
- along with regulatory issues concerning their use in disease prophylaxis and therapeutics.
- Also, the use of the state-of-the-art CRISPR/Cas9 technology in developing genetically engineered phages is elucidated.»
These topics are highly extensive, and attempting to cover them all in a single review compromises the quality of the work. Given these themes, the title does not fully align with the content of the review.
the authors need to clarify the type of review they aim to produce and establish a clear objective, such as "to compile a comprehensive list of all potential applications of bacteriophages" or "to discuss the most promising applications of phages." The chosen objective should also be explicitly stated in the Abstract.
The introduction must justify the relevance of the review. Without this, it is unclear what novel contribution the review offers. For example: "No previous review has compiled an exhaustive list of all phage applications" or "This review provides the most complete coverage of phage applications across diverse fields." It is also important to describe which reviews have previously investigated this area and to cite them accordingly
Subsequent chapters should align precisely with the review’s objective. It is recommended to organize chapters according to application areas. Sections such as "2. Advantages of Phage Therapy", "3. Disadvantages of Phage Therapy", "16. Regulatory Issues Associated with Phage Therapy", and "17. Recent Developments in Phage Engineering and Therapy" should be integrated into relevant chapters where necessary, as they currently fail to address all application areas comprehensively. The authors should also consider reducing the number of chapters by merging overlapping content.
For instance, Chapters 4, 5, 10, and 11 could logically be combined, as they complement one another but appear fragmented in the current manuscript. Similarly, Chapters 6, 7, and 8, which focus on similar applications, should be merged.
Including a table that lists pathogens, and the corresponding phages used to target them, along with references to their sources, would significantly enhance the article's value. A separate table dedicated to phage applications in cancer therapy would also be highly beneficial.
Original schematic diagrams that visually organize the application areas and sources of phages would further improve the manuscript’s uniqueness and clarity. Placing such diagrams in the introduction would engage readers and make the article more accessible.
A notable limitation of phage therapy is its low stability within a human body, necessitating the development of targeted delivery systems and protective encapsulation. Addressing the potential formulations for phage application, such as microcontainer-based approaches including polymer microcapsules and liposomes, would enhance the relevance of the article.
In «18. Conclusions», authors can enhance the value of the paper by giving their own conclusions about all the sources found. For example, they can conclude which of the described applications is the most promising based on the dates of the articles, or which of the described applications has the largest number of articles.
In its current form, the manuscript has room for improvement.
Author Response
Reviewer #3
Reviewer comment:
The review compiles a substantial amount of information but lacks a coherent narrative structure, making it difficult to read. The themes are disjointed, and locating specific information within the review is challenging.
Author rebuttal:
The manuscript has been revised to make it more organized and coherent.
In the introduction, the authors outline 10 objectives:
«The current review addresses the application of phages:
- in therapy of infectious diseases,
- neurodegenerative disorders,
- cancer
- and theragnostics
- with an insightful look into the phage display technology
- and biopanning strategy to generate engineered phages.
- The advantages
- and disadvantages of the use of phage-based therapy are also discussed
- along with regulatory issues concerning their use in disease prophylaxis and therapeutics.
- Also, the use of the state-of-the-art CRISPR/Cas9 technology in developing genetically engineered phages is elucidated.
These topics are highly extensive, and attempting to cover them all in a single review compromises the quality of the work.
Reviewer comment:
Given these themes, the title does not fully align with the content of the review.
Author rebuttal:
The title has been modified as per the content of the review.
Reviewer comment:
the authors need to clarify the type of review they aim to produce and establish a clear objective, such as "to compile a comprehensive list of all potential applications of bacteriophages" or "to discuss the most promising applications of phages." The chosen objective should also be explicitly stated in the Abstract.
Author rebuttal:
The chosen objective has been specifically stated in both the Abstract and Introduction sections as specified.
Reviewer comment:
The introduction must justify the relevance of the review. Without this, it is unclear what novel contribution the review offers. For example: "No previous review has compiled an exhaustive list of all phage applications" or "This review provides the most complete coverage of phage applications across diverse fields." It is also important to describe which reviews have previously investigated this area and to cite them accordingly.
Author rebuttal:
Previous recent literature on the topic of this review has been cited and the novelty of our current review has been addressed as per the reviewer’s valuable comment.
Reviewer comment:
Subsequent chapters should align precisely with the review’s objective. It is recommended to organize chapters according to application areas. Sections such as "2. Advantages of Phage Therapy", "3. Disadvantages of Phage Therapy", "16. Regulatory Issues Associated with Phage Therapy", and "17. Recent Developments in Phage Engineering and Therapy" should be integrated into relevant chapters where necessary, as they currently fail to address all application areas comprehensively.
Author rebuttal:
Sections including 2. Advantages of Phage Therapy", "3. Disadvantages of Phage Therapy", "16. Regulatory Issues Associated with Phage Therapy" have been presented together and the section 17. Recent Developments in Phage Engineering and Therapy" has been integrated after the Cancer therapy section where it is appropriate.
Reviewer comment:
The authors should also consider reducing the number of chapters by merging overlapping content. For instance, Chapters 4, 5, 10, and 11 could logically be combined, as they complement one another but appear fragmented in the current manuscript. Similarly, Chapters 6, 7, and 8, which focus on similar applications, should be merged.
Author rebuttal:
Chapters 4 (phage display technology), 5 (biopanning),10 (generation of mAbs using phage display technology), 6 (the section on phage cocktails) and 15 (phage engineering using CRISPR-Cas9 technology) have been combined and Chapters 7 and 8 (covering bacterial therapy) also have been merged as per the reviewer’s valuable advise. New sections on viral therapy and veterinary disease therapy have been added following which cancer therapy has been incorporated. The last section of the text addresses recent developments in phage therapy against various infectious diseases and cancer and right at the end the Conclusion section has been incorporated.
Reviewer comment:
Including a table that lists pathogens, and the corresponding phages used to target them, along with references to their sources, would significantly enhance the article's value. A separate table dedicated to phage applications in cancer therapy would also be highly beneficial.
Author rebuttal:
A table listing bacterial and viral pathogens and the respective phages used to target them along with the references to their sources has been provided, by which we have replaced the corresponding textual matter on the application of phages in the treatment of infectious diseases. Another table has been formulated showing phage applications in cancer therapy.
Reviewer comment:
Original schematic diagrams that visually organize the application areas and sources of phages would further improve the manuscript’s uniqueness and clarity. Placing such diagrams in the introduction would engage readers and make the article more accessible.
Author rebuttal:
We thank the reviewer for the valuable comment. We have included an original figure organizing the phage application areas and phage modification technologies in the Introduction section. Another figure has also been added to address the description of the phage display technology and biopanning.
Reviewer comment:
A notable limitation of phage therapy is its low stability within a human body, necessitating the development of targeted delivery systems and protective encapsulation. Addressing the potential formulations for phage application, such as microcontainer-based approaches including polymer microcapsules and liposomes, would enhance the relevance of the article.
Author rebuttal:
Comment addressed. Material has been included to cover the encapsulation of phages in liposomes and polymer microcapsules.
Reviewer comment:
In «18. Conclusions», authors can enhance the value of the paper by giving their own conclusions about all the sources found. For example, they can conclude which of the described applications is the most promising based on the dates of the articles, or which of the described applications has the largest number of articles.
Author rebuttal:
This comment has been duly addressed in the Conclusion section.
In its current form, the manuscript has room for improvement.

Reviewer 4 Report
Comments and Suggestions for Authors
The Authors of the manuscript have tried to examine the use of bacteriophages in directed therapy and some corresponding aspects, such as the selection of phages, targets, targeting ligands, phage engineering and safety of phage-based therapeutic approaches. These topics are one of hot spots of directed therapies in the fields of microbial infections, cancer therapy and diagnostics, including theranostics, and immunotherapy. Despite of the topic importance, the manuscript may not attract much attention because of its poor content, text and figure organization, some drawbacks and mistakes. The abstract is well written, however, some topics declared in the abstract are not well described in the full text, e.g. the influence of the "copy number of ligands" on the transfection effeciency; "high level of stability as well as resistance of phages to various environmental conditions" (only thermostability is mentioned); VLP discussion; "phage display technology in generating monoclonal antibodies" (not only reference but an explanation of the technology).
Regarding the text as a whole, it contains many repeats of the facts mentioned iin different sections and different parts of one and the same section. E.g.,1) "... only completely lytic phages are employed for therapeutic purposes"; 2) the technology of phage dislplay and biopanning strategies as well as CRISPR Cas9 editing correspond closely to the phage engineering and should be discussed together with this topic and not in separate sections, otherwise one can see many text repeats; 3) the note that phages represent foreign particles, which can be immunogenic is repeated at list twice; 4) Folate-conjugated M13 phage with doxorubicin encapsulated in PCL-P2VP is also mentioned twice.
The major part of the whole text represents a sum of facts (some of which are taken from outdated publications) with no summarizing and analysis of them. The authors do not discuss and cite the publications from the group of Dr. Peabody, who is one of the main persons in MS2-VLPs research. Instead of older references 56, 118, 119 much more recent publications should be discussed and cited: 1) DOI: 10.1016/S1473-3099(18)30482-1; 2) DOI: 10.1007/s10517-024-06209-6; 3) DOI:10.1007/s10517-024-06225-6.
Figures 1A and 1B, 4(A-D) should be presented as separately numbered figures located close to the text they correspond to.
"A novel peptidase" CHAPK is a member of the well-known group of phage peptidases endolysins. Some of these enzymes are used in therapeutic preparations.
Comments on the Quality of English LanguageThe text of the manuscript requires English editing, since some phrases cannot be well understood, some have grammatical constructs not typical for the English language. It is typical for the text sections 5, 6, 7, 12.
Author Response
Reviewer #4
The Authors of the manuscript have tried to examine the use of bacteriophages in directed therapy and some corresponding aspects, such as the selection of phages, targets, targeting ligands, phage engineering and safety of phage-based therapeutic approaches. These topics are one of hot spots of directed therapies in the fields of microbial infections, cancer therapy and diagnostics, including theranostics, and immunotherapy.
Reviewer comment:
Despite of the topic importance, the manuscript may not attract much attention because of its poor content, text and figure organization, some drawbacks and mistakes.
Author rebuttal:
Several sections have been reorganized, repeated / redundant sentences have been removed, and additional figures and tables have been added to enhance the manuscript.
Reviewer comment:
The abstract is well written, however, some topics declared in the abstract are not well described in the full text, e.g. the influence of the "copy number of ligands" on the transfection effeciency; "high level of stability as well as resistance of phages to various environmental conditions" (only thermostability is mentioned); VLP discussion; "phage display technology in generating monoclonal antibodies" (not only reference but an explanation of the technology).
Author rebuttal:
The abstract has been reformulated as per Reviewer #2’s comments and therefore the sentences containing “the influence of the copy number of ligands on the transfection efficiency” and “The high level of stability as well as resistance to environmental conditions have enabled the development of VLPs” have been removed from the abstract. Phage stability to various environmental conditions has been duly discussed under a separate heading in the text. Explanation of phage display technology in generating monoclonal antibodies has also been addressed.
Reviewer comment:
Regarding the text as a whole, it contains many repeats of the facts mentioned iin different sections and different parts of one and the same section. E.g.,1) "... only completely lytic phages are employed for therapeutic purposes"; 2) the technology of phage dislplay and biopanning strategies as well as CRISPR Cas9 editing correspond closely to the phage engineering and should be discussed together with this topic and not in separate sections, otherwise one can see many text repeats; 3) the note that phages represent foreign particles, which can be immunogenic is repeated at list twice; 4) Folate-conjugated M13 phage with doxorubicin encapsulated in PCL-P2VP is also mentioned twice.
Author rebuttal:
The repeated sentences stated as above have been deleted and the sections discussing phage display technology, biopanning strategies, production of mAbs using phage display technology as well as phage engineering using CRISPR-Cas9 technology have been combined as per the reviewer’s valuable advise. Also, sections containing phage therapy in treating bacterial, viral, veterinary infections, cancer and neurotherapy have been put together.
Reviewer comment:
The major part of the whole text represents a sum of facts (some of which are taken from outdated publications) with no summarizing and analysis of them. The authors do not discuss and cite the publications from the group of Dr. Peabody, who is one of the main persons in MS2-VLPs research.
Author rebuttal:
The work of Dr. Peabody on the use of MS2 VLPs to identify pathogen epitopes displayed on their surface that are capable of binding a repertoire of antibodies against viral diseases in humans using an antigen fragment library specific to the respective virus combined with deep sequence coupled biopanning has been cited and discussed.
Reviewer comment:
Instead of older references 56, 118, 119 much more recent publications should be discussed and cited: 1) DOI: 10.1016/S1473-3099(18)30482-1; 2) DOI: 10.1007/s10517-024-06209-6; 3) DOI:10.1007/s10517-024-06225-6.
Author rebuttal:
The older references have been replaced with new ones and discussed accordingly as specified by the reviewer.
Reviewer comment:
Figures 1A and 1B, 4(A-D) should be presented as separately numbered figures located close to the text they correspond to.
Author rebuttal:
Figures 1A and 1B and Figure 4 (A-D) have been shown as separately numbered figures at places close to their respective text.
Reviewer comment:
"A novel peptidase" CHAPK is a member of the well-known group of phage peptidases endolysins. Some of these enzymes are used in therapeutic preparations.
Author rebuttal:
The role of bacteriophage CHAPK in therapeutic applications is discussed as per the reviewer’s advise.
Reviewer comment:
Comments on the Quality of English Language
The text of the manuscript requires English editing, since some phrases cannot be well understood, some have grammatical constructs not typical for the English language. It is typical for the text sections 5, 6, 7, 12.
Author rebuttal:
The necessary language corrections have been incorporated.

Round 2
Reviewer 2 Report
Comments and Suggestions for Authors
Authors resolves all confused points. The manuscript could be considered by the last decision of Editor before publication.
Author Response
We thank the reviewer for approving the latest version of our manuscript. We are grateful to the reviewer for all the valuable advise.
Reviewer 3 Report
Comments and Suggestions for Authors
The article (bioengineering-3527232) has been greatly improved since the previous revision. The authors have made changes, and the text seems more structured.
The authors have changed the title of the article (Bacteriophages as Targeted Therapeutic Vehicles: Challenges and Opportunities"). The new title seems more appropriate to me. However, I draw the authors' attention to the fact that there is now a typographical error in the title (an inverted comma at the end).
The authors have made all the suggested changes, including outlining the purpose of the paper and adding several Figures and Tables. A large number of new references have been added. This has a positive impact on the quality of the work. But I can recommend the authors edit Table 3. The Reference column could be made narrower, and the other columns could be made wider.
The authors have added an interesting section on encapsulation of bacteriophages, which broadens the readership.
The revised version of the Conclusion is in line with the content of the article.
In the present form, the article has increased scientific value and may be of interest to the scientific community. I can recommend this version of the article for publication.
Author Response
Reviewer #3:
The article (bioengineering-3527232) has been greatly improved since the previous revision. The authors have made changes, and the text seems more structured.
Reviewer comment:
The authors have changed the title of the article (Bacteriophages as Targeted Therapeutic Vehicles: Challenges and Opportunities"). The new title seems more appropriate to me. However, I draw the authors' attention to the fact that there is now a typographical error in the title (an inverted comma at the end).
Author rebuttal:
This typo has been corrected.
Reviewer comment:
The authors have made all the suggested changes, including outlining the purpose of the paper and adding several Figures and Tables. A large number of new references have been added. This has a positive impact on the quality of the work. But I can recommend the authors edit Table 3. The Reference column could be made narrower, and the other columns could be made wider.
Author rebuttal:
Table 3 reference column has been made narrower and the other columns wider as per the reviewer’s valuable advise.
The authors have added an interesting section on encapsulation of bacteriophages, which broadens the readership. The revised version of the Conclusion is in line with the content of the article. In the present form, the article has increased scientific value and may be of interest to the scientific community. I can recommend this version of the article for publication.
Reviewer 4 Report
Comments and Suggestions for Authors
My decision is that it can be accepted after minor revisions:
The first version of the manuscript has been substantially improved. The only notes are:
1) The legends to the figures 5 and 6 are too long, they should be reduced maybe by transfering a part of the information to the text.
2) The Photodynamic therapy - PDT - abbreviation (page 38, beginnings of the 2nd and 3rd paragraphs) should be used once in the text, no need to repeat it.
Author Response
Reviewer #4
My decision is that it can be accepted after minor revisions: The first version of the manuscript has been substantially improved.
Reviewer comment:
The only notes are:
1) The legends to the figures 5 and 6 are too long, they should be reduced maybe by transferring a part of the information to the text.
Author rebuttal:
The legends of Figure 5 and 6 have been duly reduced as per the reviewer’s valuable advise.
Reviewer comment:
2) The Photodynamic therapy - PDT - abbreviation (page 38, beginnings of the 2nd and 3rd paragraphs) should be used once in the text, no need to repeat it.
Author rebuttal:
The terms photothermal therapy (PTT) and photodynamic therapy (PDT) have been mentioned as PTT and PDT respectively at their first mention on page 4 following which they are referred as PTT and PDT.